# HyperHuman: Hyper-Realistic Human Generation with Latent Structural Diffusion

Xian Liu[1,2*]   Jian Ren[1†]   Aliaksandr Siarohin[1]   Ivan Skorokhodov[1]   Yanyu Li[1]
Dahua Lin[2]   Xihui Liu[3]   Ziwei Liu[4]   Sergey Tulyakov[1]
[1]Snap Inc.   [2]CUHK   [3]HKU   [4]NTU
Project Page: https://snap-research.github.io/HyperHuman

## Abstract

Despite significant advances in large-scale text-to-image models, achieving hyper-realistic human image generation remains a desirable yet unsolved task. Existing models like Stable Diffusion and DALL·E 2 tend to generate human images with incoherent parts or unnatural poses. To tackle these challenges, our key insight is that human image is inherently structural over multiple granularities, from the coarse-level body skeleton to the fine-grained spatial geometry. Therefore, capturing such correlations between the explicit appearance and latent structure in one model is essential to generate coherent and natural human images. To this end, we propose a unified framework, **HyperHuman**, that generates in-the-wild human images of high realism and diverse layouts. Specifically, **1)** we first build a large-scale human-centric dataset, named *HumanVerse*, which consists of $340M$ images with comprehensive annotations like human pose, depth, and surface-normal. **2)** Next, we propose a *Latent Structural Diffusion Model* that simultaneously denoises the depth and surface-normal along with the synthesized RGB image. Our model enforces the joint learning of image appearance, spatial relationship, and geometry in a unified network, where each branch in the model complements to each other with both structural awareness and textural richness. **3)** Finally, to further boost the visual quality, we propose a *Structure-Guided Refiner* to compose the predicted conditions for more detailed generation of higher resolution. Extensive experiments demonstrate that our framework yields the state-of-the-art performance, generating hyper-realistic human images under diverse scenarios.

## 1   Introduction

Generating hyper-realistic human images from user conditions, *e.g.,* text and pose, is of great importance to various applications, such as image animation (Liu et al., 2019) and virtual try-on (Wang et al., 2018). To this end, many efforts explore the task of controllable human image generation. Early methods either resort to variational auto-encoders (VAEs) in a reconstruction manner (Ren et al., 2020), or improve the realism by generative adversarial networks (GANs) (Siarohin et al., 2019). Though some of them create high-quality images (Zhang et al., 2022; Jiang et al., 2022), the unstable training and limited model capacity confine them to small datasets of low diversity. Recent emergence of diffusion models (DMs) (Ho et al., 2020) has set a new paradigm for realistic synthesis and become the predominant architecture in Generative AI (Dhariwal & Nichol, 2021). Nevertheless, the exemplar text-to-image (T2I) models like Stable Diffusion (Rombach et al., 2022) and DALL·E 2 (Ramesh et al., 2022) still struggle to create human images with coherent anatomy, *e.g.,* arms and legs, and natural poses. The main reason lies in that human is articulated with non-rigid deformations, requiring structural information that can hardly be depicted by text prompts.

To enable structural control for image generation, recent works like ControlNet (Zhang & Agrawala, 2023) and T2I-Adapter (Mou et al., 2023) introduce a learnable branch to modulate the pre-trained DMs, *e.g.,* Stable Diffusion, in a plug-and-play manner. However, these approaches suffer from the

---

*Work done during an internship at Snap Inc. Email: alvinliu@ie.cuhk.edu.hk
†Corresponding author: jren@snapchat.com.

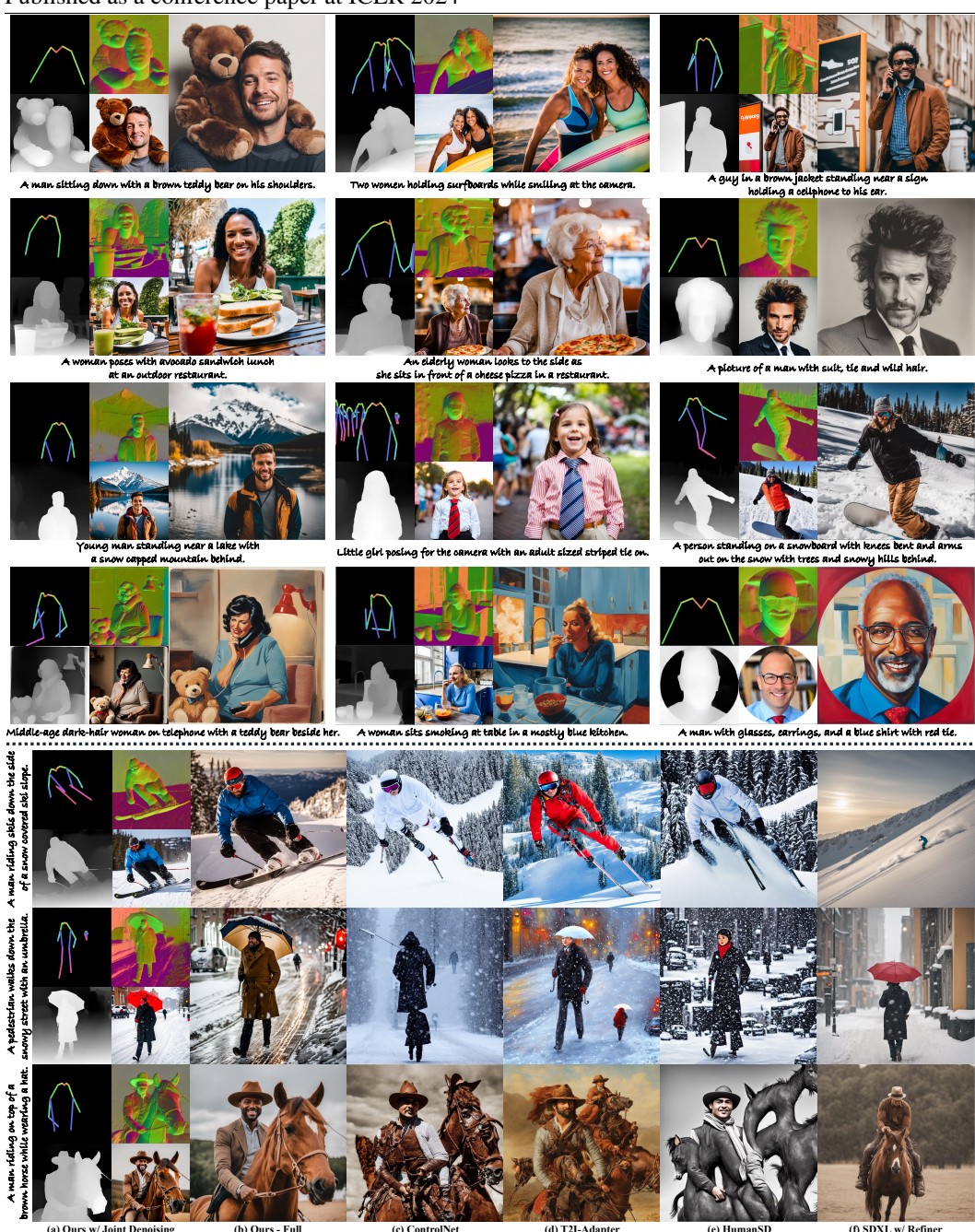

Figure 1: **Example Results and Visual Comparison.** *Top:* The proposed **HyperHuman** simultaneously generates the coarse RGB, depth, normal, and high-resolution images conditioned on text and skeleton. Both photo-realistic images and stylistic renderings can be created. *Bottom:* We compare with recent T2I models, showing better realism, quality, diversity, and controllability. Note that in each $2 \times 2$ grid (**left**), the upper-left is *input* skeleton, while the others are jointly denoised normal, depth, and coarse RGB of $512 \times 512$. With full model, we synthesize images up to $1024 \times 1024$ (**right**). Please refer to Sec. A.15, A.16 for more comparison and results. **Best viewed zoom in**.

feature discrepancy between the main and auxiliary branches, leading to inconsistency between the control signals (*e.g.,* pose maps) and the generated images. To address the issue, HumanSD (Ju et al., 2023b) proposes to directly input body skeleton into the diffusion U-Net by channel-wise concatenation. However, it is confined to generating artistic style images of limited diversity. Besides, human images are synthesized only with pose control, while other structural information like depth maps and surface-normal maps are not considered. In a nutshell, previous studies either take a singular control signal as input condition, or treat different control signals separately as independent guidance, instead of modeling the multi-level correlations between human appearance and different types of structural information. Realistic human generation with coherent structure remains unsolved.

In this paper, we propose a unified framework **HyperHuman** to generate in-the-wild human images of high realism and diverse layouts. The key insight is that *human image is inherently structural over multiple granularities, from the coarse-level body skeleton to fine-grained spatial geometry*. Therefore, capturing such correlations between the explicit appearance and latent structure in one model is essential to generate coherent and natural human images. Specifically, we first establish a large-scale human-centric dataset called *HumanVerse* that contains 340M in-the-wild human images of high quality and diversity. It has comprehensive annotations, such as the coarse-level body skeletons, the fine-grained depth and surface-normal maps, and the high-level image captions and attributes. Based on this, two modules are designed for hyper-realistic controllable human image generation. In *Latent Structural Diffusion Model*, we augment the pre-trained diffusion backbone to simultaneously denoise the RGB, depth, and normal. Appropriate network layers are chosen to be replicated as structural expert branches, so that the model can both handle input/output of different domains, and guarantee the spatial alignment among the denoised textures and structures. Thanks to such dedicated design, the image appearance, spatial relationship, and geometry are jointly modeled within a unified network, where each branch is complementary to each other with both structural awareness and textural richness. To generate monotonous depth and surface-normal that have similar values in local regions, we utilize an improved noise schedule to eliminate low-frequency information leakage. The same timestep is sampled for each branch to achieve better learning and feature fusion. With the spatially-aligned structure maps, in *Structure-Guided Refiner*, we compose the predicted conditions for detailed generation of high resolution. Moreover, we design a robust conditioning scheme to mitigate the effect of error accumulation in our two-stage generation pipeline.

To summarize, our main contributions are three-fold: **1)** We propose a novel **HyperHuman** framework for in-the-wild controllable human image generation of high realism. A large-scale human-centric dataset *HumanVerse* is curated with comprehensive annotations like human pose, depth, and surface normal. As one of the earliest attempts in human generation foundation model, we hope to benefit future research. **2)** We propose the *Latent Structural Diffusion Model* to jointly capture the image appearance, spatial relationship, and geometry in a unified framework. The *Structure-Guided Refiner* is further devised to compose the predicted conditions for generation of better visual quality and higher resolution. **3)** Extensive experiments demonstrate that our **HyperHuman** yields the state-of-the-art performance, generating hyper-realistic human images under diverse scenarios.

## 2 RELATED WORK

**Text-to-Image Diffusion Models.** Text-to-image (T2I) generation, the endeavor to synthesize high-fidelity images from natural language descriptions, has made remarkable strides in recent years. Distinguished by the superior scalability and stable training, diffusion-based T2I models have eclipsed conventional GANs in terms of performance (Dhariwal & Nichol, 2021), becoming the predominant choice in generation (Nichol et al., 2021; Saharia et al., 2022; Balaji et al., 2022; Li et al., 2023). By formulating the generation as an iterative denoising process (Ho et al., 2020), exemplar works like Stable Diffusion (Rombach et al., 2022) and DALL·E 2 (Ramesh et al., 2022) demonstrate unprecedented quality. Despite this, they mostly fail to create high-fidelity humans. One main reason is that existing models lack inherent structural awareness for human, making them even struggle to generate human of reasonable anatomy, *e.g.,* correct number of arms and legs. To this end, our proposed approach explicitly models human structures within the latent space of diffusion model.

**Controllable Human Image Generation.** Traditional approaches for controllable human generation can be categorized into GAN-based (Zhu et al., 2017; Siarohin et al., 2019) and VAE-based (Ren et al., 2020; Yang et al., 2021), where the reference image and conditions are taken as input. To facilitate user-friendly applications, recent studies explore text prompts as generation guidance (Roy et al., 2022; Jiang et al., 2022), yet are confined to simple pose or style descriptions. The most relevant works that enable open-vocabulary pose-guided controllable human synthesis are ControlNet (Zhang & Agrawala, 2023), T2I-Adapter (Mou et al., 2023), and HumanSD (Ju et al., 2023b). However, they either suffer from inadequate pose control, or are confined to artistic styles of limited diversity. Besides, most previous studies merely take pose as input, while ignoring the multi-level correlations between human appearance and different types of structural information. In this work, we propose to incorporate structural awareness from coarse-level skeleton to fine-grained depth and surface-normal by joint denoising with expert branch, thus simultaneously capturing both the explicit appearance and latent structure in a unified framework for realistic human image synthesis.

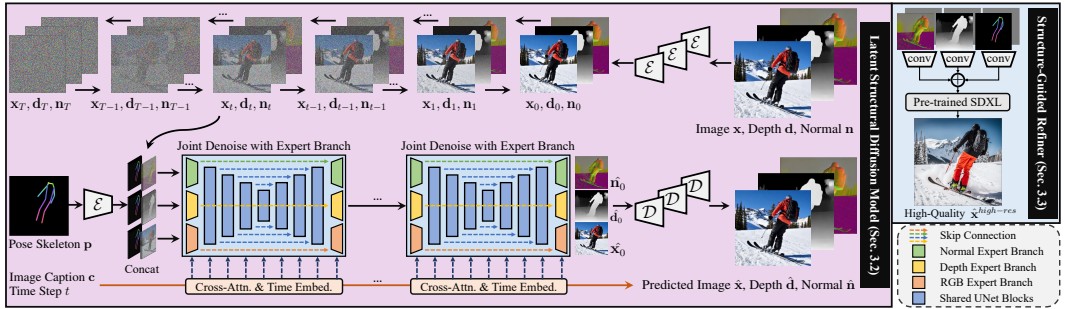

Figure 2: **Overview of HyperHuman Framework.** In *Latent Structural Diffusion Model* (purple), the image **x**, depth **d**, and surface-normal **n** are jointly denoised conditioning on caption **c** and pose skeleton **p**. For the notation simplicity, we denote pixel-/latent-space targets with the same variable. In *Structure-Guided Refiner* (blue), we compose the predicted conditions for higher-resolution generation. Note that the grey images refer to randomly dropout conditions for more robust training.

**Datasets for Human Image Generation.** Large datasets are crucial for image generation. Existing human-centric collections are mainly confronted with following drawbacks: **1)** Low-resolution of poor quality. For example, Market-1501 (Zheng et al., 2015) contains noisy pedestrian images of resolution $128 \times 64$, and VITON (Han et al., 2018) has human-clothing pairs of $256 \times 192$, which are inadequate for training high-definition models. **2)** Limited diversity of certain domain. For example, SHHQ (Fu et al., 2022) is mostly composed of full-body humans with clean background, and DeepFashion (Liu et al., 2016) focuses on fashion images of little pose variations. **3)** Insufficient dataset scale, where LIP (Gong et al., 2017) and Human-Art (Ju et al., 2023a) only contain 50K samples. Furthermore, none of the existing datasets contain rich annotations, which typically label a singular aspect of images. In this work, we take a step further by curating in-the-wild *HumanVerse* dataset with comprehensive annotations like human pose, depth map, and surface-normal map.

## 3 OUR APPROACH

We present **HyperHuman** that generates in-the-wild human images of high realism and diverse layouts. The overall framework is illustrated in Fig. 2. To make the content self-contained and narration clearer, we first introduce some pre-requisites of diffusion models and the problem setting in Sec. 3.1. Then, we present the *Latent Structural Diffusion Model* which simultaneously denoises the depth, surface-normal along with the RGB image. The explicit appearance and latent structure are thus jointly learned in a unified model (Sec. 3.2). Finally, we elaborate the *Structure-Guided Refiner* to compose the predicted conditions for detailed generation of higher resolution in Sec. 3.3.

### 3.1 PRELIMINARIES AND PROBLEM SETTING

**Diffusion Probabilistic Models** define a forward diffusion process to gradually convert the sample **x** from a real data distribution $p_{\text{data}}(\mathbf{x})$ into a noisy version, and learn the reverse generation process in an iterative denoising manner (Sohl-Dickstein et al., 2015; Song et al., 2020b). During the sampling stage, the model can transform Gaussian noise of normal distribution to real samples step-by-step. The denoising network $\hat{\boldsymbol{\epsilon}}_{\boldsymbol{\theta}}(\cdot)$ estimates the additive Gaussian noise, which is typically structured as a UNet (Ronneberger et al., 2015) to minimize the ensemble of mean-squared error (Ho et al., 2020):

$$\min_{\boldsymbol{\theta}} \ \mathbb{E}_{\mathbf{x},\mathbf{c},\boldsymbol{\epsilon},t} \ \left[ w_t \|\hat{\boldsymbol{\epsilon}}_{\boldsymbol{\theta}}(\alpha_t \mathbf{x} + \sigma_t \boldsymbol{\epsilon}; \mathbf{c}) - \boldsymbol{\epsilon}\|_2^2 \right], \tag{1}$$

where $\mathbf{x}, \mathbf{c} \sim p_{\text{data}}$ are the sample-condition pairs from the training distribution; $\boldsymbol{\epsilon} \sim \mathcal{N}(\mathbf{0}, \mathbf{I})$ is the ground-truth noise; $t \sim \mathcal{U}[1, T]$ is the time-step and $T$ is the training step number; $\alpha_t$, $\sigma_t$, and $w_t$ are the terms that control the noise schedule and sample quality decided by the diffusion sampler.

**Latent Diffusion Model & Stable Diffusion.** The widely-used latent diffusion model (LDM), with its improved version Stable Diffusion (Rombach et al., 2022), performs the denoising process in a separate latent space to reduce the computational cost. Specifically, a pre-trained VAE (Esser et al., 2021) first encodes the image **x** to latent embedding $\mathbf{z} = \mathcal{E}(\mathbf{x})$ for DM training. At the inference

stage, we can reconstruct the generated image through the decoder $\hat{\mathbf{x}} = \mathcal{D}(\hat{\mathbf{z}})$. Such design enables the SD to scale up to broader datasets and larger model size, advancing from the *SD 1.x & 2.x* series to *SDXL* of heavier backbone on higher resolution (Podell et al., 2023). In this work, we extend *SD 2.0* to *Latent Structural Diffusion Model* for efficient capturing of explicit appearance and latent structure, while the *Structure-Guided Refiner* is built on *SDXL 1.0* for more pleasing visual quality.

**Problem Setting for Controllable Human Generation.** Given a collection of $N$ human images $\mathbf{x}$ with their captions $\mathbf{c}$, we annotate the depth $\mathbf{d}$, surface-normal $\mathbf{n}$, and pose skeleton $\mathbf{p}$ for each sample (details elaborated in Sec. 4). The training dataset can be denoted as $\{\mathbf{x}_i, \mathbf{c}_i, \mathbf{d}_i, \mathbf{n}_i, \mathbf{p}_i\}_{i=1}^N$. In the first-stage *Latent Structural Diffusion Model* $\mathcal{G}_1$, we estimate the RGB image $\hat{\mathbf{x}}$, depth $\hat{\mathbf{d}}$, and surface-normal $\hat{\mathbf{n}}$ conditioned on the caption $\mathbf{c}$ and skeleton $\mathbf{p}$. In the second-stage *Structure-Guided Refiner* $\mathcal{G}_2$, the predicted structures of $\hat{\mathbf{d}}$ and $\hat{\mathbf{n}}$ further serve as guidance for the generation of higher-resolution results $\hat{\mathbf{x}}^{high\text{-}res}$. The training setting for our pipeline can be formulated as:

$$\hat{\mathbf{x}}, \hat{\mathbf{d}}, \hat{\mathbf{n}} = \mathcal{G}_1(\mathbf{c}, \mathbf{p}), \qquad \hat{\mathbf{x}}^{high\text{-}res} = \mathcal{G}_2(\mathbf{c}, \mathbf{p}, \hat{\mathbf{d}}, \hat{\mathbf{n}}). \tag{2}$$

During inference, only the text prompt and body skeleton are needed to synthesize well-aligned RGB image, depth, and surface-normal. Note that the users are free to substitute their own depth and surface-normal conditions to $\mathcal{G}_2$ if applicable, enabling more flexible and controllable generation.

## 3.2 LATENT STRUCTURAL DIFFUSION MODEL

To incorporate the body skeletons for pose control, the simplest way is by feature residual (Mou et al., 2023) or input concatenation (Ju et al., 2023b). However, three problems remain: **1)** The sparse keypoints only depict the coarse human structure, while the fine-grained geometry and foreground-background relationship are ignored. Besides, the naive DM training is merely supervised by RGB signals, which fails to capture the inherent structural information. **2)** The image RGB and structure representations are spatially aligned but substantially different in latent space. How to jointly model them remains challenging. **3)** In contrast to the colorful RGB images, the structure maps are mostly monotonous with similar values in local regions, which are hard to learn by DMs (Lin et al., 2023).

**Unified Model for Simultaneous Denoising.** Our solution to the first problem is to simultaneously denoise the depth and surface-normal along with the synthesized RGB image. We choose them as additional learning targets due to two reasons: **1)** Depth and normal can be easily annotated for large-scale dataset, which are also used in recent controllable T2I generation (Zhang & Agrawala, 2023). **2)** As two commonly-used structural guidance, they complement the spatial relationship and geometry information, where the depth (Deng et al., 2022), normal (Wang et al., 2022), or both (Yu et al., 2022b) are proven beneficial in recent 3D studies. To this end, a naive method is to train three separate networks to denoise the RGB, depth, and normal individually. But the spatial alignment between them is hard to preserve. Therefore, we propose to capture the joint distribution in a unified model by simultaneous denoising, which can be trained with simplified objective (Ho et al., 2020):

$$\mathcal{L}^{\epsilon\text{-}pred} = \mathbb{E}_{\mathbf{x},\mathbf{d},\mathbf{n},\mathbf{c},\boldsymbol{\epsilon},t} \big[ \underbrace{||\hat{\boldsymbol{\epsilon}}_{\boldsymbol{\theta}}(\mathbf{x}_{t_{\mathbf{x}}}; \mathbf{c}, \mathbf{p}) - \boldsymbol{\epsilon}_{\mathbf{x}}||_2^2}_{\text{denoise image } \mathbf{x}} + \underbrace{||\hat{\boldsymbol{\epsilon}}_{\boldsymbol{\theta}}(\mathbf{d}_{t_{\mathbf{d}}}; \mathbf{c}, \mathbf{p}) - \boldsymbol{\epsilon}_{\mathbf{d}}||_2^2}_{\text{denoise depth } \mathbf{d}} + \underbrace{||\hat{\boldsymbol{\epsilon}}_{\boldsymbol{\theta}}(\mathbf{n}_{t_{\mathbf{n}}}; \mathbf{c}, \mathbf{p}) - \boldsymbol{\epsilon}_{\mathbf{n}}||_2^2}_{\text{denoise normal } \mathbf{n}} \big], \tag{3}$$

where $\boldsymbol{\epsilon}_{\mathbf{x}}$, $\boldsymbol{\epsilon}_{\mathbf{d}}$, and $\boldsymbol{\epsilon}_{\mathbf{n}} \sim \mathcal{N}(\mathbf{0}, \mathbf{I})$ are three independently sampled Gaussian noise (shortened as $\boldsymbol{\epsilon}$ in expectation for conciseness) for the RGB, depth, and normal, respectively; $\mathbf{x}_{t_{\mathbf{x}}} = \alpha_{t_{\mathbf{x}}}\mathbf{x} + \sigma_{t_{\mathbf{x}}}\boldsymbol{\epsilon}_{\mathbf{x}}$, $\mathbf{d}_{t_{\mathbf{d}}} = \alpha_{t_{\mathbf{d}}}\mathbf{d} + \sigma_{t_{\mathbf{d}}}\boldsymbol{\epsilon}_{\mathbf{d}}$, and $\mathbf{n}_{t_{\mathbf{n}}} = \alpha_{t_{\mathbf{n}}}\mathbf{n} + \sigma_{t_{\mathbf{n}}}\boldsymbol{\epsilon}_{\mathbf{n}}$ are the noised feature maps of three learning targets; $t_{\mathbf{x}}$, $t_{\mathbf{d}}$, and $t_{\mathbf{n}} \sim \mathcal{U}[1, T]$ are the sampled time-steps that control the scale of added Gaussian noise.

**Structural Expert Branches with Shared Backbone.** The diffusion UNet contains down-sample, middle, and up-sample blocks, which are interleaved with convolution and self-/cross-attention layers. In particular, the *DownBlock*s compress input noisy latent to the hidden states of lower resolution, while the *UpBlock*s conversely upscale intermediate features to the predicted noise. Therefore, the most intuitive manner is to replicate the first several *DownBlock*s and the last several *UpBlock*s for each expert branch, which are the most neighboring layers to the input and output. In this way, each expert branch gradually maps input noisy latent of different domains (*i.e.,* $\mathbf{x}_{t_{\mathbf{x}}}$, $\mathbf{d}_{t_{\mathbf{d}}}$, and $\mathbf{n}_{t_{\mathbf{n}}}$) to similar distribution for feature fusion. Then, after a series of shared modules, the same feature is distributed to each expert branch to output noises (*i.e.,* $\boldsymbol{\epsilon}_{\mathbf{x}}$, $\boldsymbol{\epsilon}_{\mathbf{d}}$, and $\boldsymbol{\epsilon}_{\mathbf{n}}$) for spatially-aligned results.

Furthermore, we find that the number of shared modules can trade-off between the spatial alignment and distribution learning: On the one hand, more shared layers guarantee the more similar features

of final output, leading to the paired texture and structure corresponding to the same image. On the other hand, the RGB, depth, and normal can be treated as different views of the same image, where predicting them from the same feature resembles an image-to-image translation task in essence. Empirically, we find the optimal design to replicate the *conv_in*, first *DownBlock*, last *UpBlock*, and *conv_out* for each expert branch, where each branch's skip-connections are maintained separately (as depicted in Fig. 2). This yields both the spatial alignment and joint capture of image texture and structure. Note that such design is not limited to three targets, but can generalize to arbitrary number of paired distributions by simply involving more branches with little computation overhead.

**Noise Schedule for Joint Learning.** A problem arises when we inspect the distribution of depth and surface-normal: After annotated by off-the-shelf estimators, they are regularized to certain data range with similar values in local regions, *e.g.,* $[0, 1]$ for depth and unit vector for surface-normal. Such monotonous images may leak low-frequency signals like the mean of each channel during training. Besides, their latent distributions are divergent from that of RGB space, making them hard to exploit common noise schedules (Lin et al., 2023) and diffusion prior. Motivated by this, we first normalize the depth and normal latent features to the similar distribution of RGB latent, so that the pre-trained denoising knowledge can be adaptively used. The zero terminal SNR ($\alpha_T = 0, \sigma_T = 1$) is further enforced to eliminate structure map's low-frequency information. Another question is how to sample time-step $t$ for each branch. An alternative is to perturb the data of different modalities with different levels (Bao et al., 2023), which samples different $t$ for each target as in Eq. 3. However, as we aim to jointly model RGB, depth, and normal, such strategy only gives $10^{-9}$ probability to sample each perturbation situation (given total steps $T = 1000$), which is too *sparse* to obtain good results. In contrast, we propose to *densely* sample with the same time-step $t$ for all the targets, so that the sampling sparsity and learning difficulty will not increase even when we learn more modalities. With the same noise level for each structural expert branch, intermediate features follow the similar distribution when they fuse in the shared backbone, which could better complement to each others. Finally, we utilize the **v**-prediction (Salimans & Ho, 2022) learning target as network objective:

$$\mathcal{L}^{\mathbf{v}\text{-}pred} = \mathbb{E}_{\mathbf{x},\mathbf{d},\mathbf{n},\mathbf{c},\mathbf{p},\mathbf{v},t} \left[ ||\hat{\mathbf{v}}_{\boldsymbol{\theta}}(\mathbf{x}_t; \mathbf{c}, \mathbf{p}) - \mathbf{v}_t^{\mathbf{x}}||_2^2 + ||\hat{\mathbf{v}}_{\boldsymbol{\theta}}(\mathbf{d}_t; \mathbf{c}, \mathbf{p}) - \mathbf{v}_t^{\mathbf{d}}||_2^2 + ||\hat{\mathbf{v}}_{\boldsymbol{\theta}}(\mathbf{n}_t; \mathbf{c}, \mathbf{p}) - \mathbf{v}_t^{\mathbf{n}}||_2^2 \right], \quad (4)$$

where $\mathbf{v}_t^{\mathbf{x}} = \alpha_t \boldsymbol{\epsilon}_{\mathbf{x}} - \sigma_t \mathbf{x}$, $\mathbf{v}_t^{\mathbf{d}} = \alpha_t \boldsymbol{\epsilon}_{\mathbf{d}} - \sigma_t \mathbf{d}$, and $\mathbf{v}_t^{\mathbf{n}} = \alpha_t \boldsymbol{\epsilon}_{\mathbf{n}} - \sigma_t \mathbf{n}$ are the **v**-prediction learning targets at time-step $t$ for the RGB, depth, and normal, respectively. Overall, the unified simultaneous denoising network $\hat{\mathbf{v}}_{\boldsymbol{\theta}}$ with the structural expert branches, accompanied by the improved noise schedule and time-step sampling strategy give the first-stage *Latent Structural Diffusion Model* $\mathcal{G}_1$.

## 3.3 STRUCTURE-GUIDED REFINER

**Compose Structures for Controllable Generation.** With the unified latent structural diffusion model, spatially-aligned conditions of depth and surface-normal can be predicted. We then learn a refiner network to render high-quality image $\hat{\mathbf{x}}^{high\text{-}res}$ by composing multi-conditions of caption $\mathbf{c}$, pose skeleton $\mathbf{p}$, the predicted depth $\hat{\mathbf{d}}$, and the predicted surface-normal $\hat{\mathbf{n}}$. In contrast to Zhang & Agrawala (2023) and Mou et al. (2023) that can only handle a singular condition per run, we propose to unify multiple control signals at the training phase. Specifically, we first project each condition from input image size (*e.g.,* $1024 \times 1024$) to feature space vector that matches the size of *SDXL* (*e.g.,* $128 \times 128$). Each condition is encoded via a light-weight embedder of four stacked convolutional layers with $4 \times 4$ kernels, $2 \times 2$ strides, and ReLU activation. Next, the embeddings from each branch are summed up coordinate-wise and further feed into the trainable copy of *SDXL Encoder Block*s. Since involving more conditions only incurs negligible computational overhead of a tiny encoder network, our method can be trivially extended to new structural conditions. Although a recent work also incorporates multiple conditions in one model (Huang et al., 2023), they have to re-train the whole backbone, making the training cost unaffordable when scaling up to high resolution.

**Random Dropout for Robust Conditioning.** Since the predicted depth and surface-normal conditions from $\mathcal{G}_1$ may contain artifacts, a potential issue for such two-stage pipeline is the error accumulation, which typically leads to the train-test performance gap. To solve this problem, we propose to dropout structural maps for robust conditioning. In particular, we randomly mask out any of the control signals, such as replace text prompt with empty string, or substitute the structural maps with zero-value images. In this way, the model will not solely rely on a single guidance for synthesis, thus balancing the impact of each condition robustly. To sum up, the structure-composing refiner network with robust conditioning scheme constitute the second-stage *Structure-Guided Refiner* $\mathcal{G}_2$.

## 4 HUMANVERSE DATASET

Large-scale datasets with high quality samples, rich annotations, and diverse distribution are crucial for image generation tasks (Schuhmann et al., 2022; Podell et al., 2023), especially in the human domain (Liu et al., 2016; Fu et al., 2022). To facilitate controllable human generation of high-fidelity, we establish a comprehensive human dataset with extensive annotations named *HumanVerse*. Please kindly refer to Appendix A.17 for more details about the dataset and annotation resources we use.

**Dataset Preprocessing.** We curate from two principled datasets: LAION-2B-en (Schuhmann et al., 2022) and COYO-700M (Byeon et al., 2022). To isolate human images, we employ YOLOS (Fang et al., 2021) for human detection. Specifically, only those images containing 1 to 3 human bounding boxes are retained, where people should be visible with an area ratio exceeding $15\%$. We further rule out samples of poor aesthetics ($< 4.5$) or low resolution ($< 200 \times 200$). This yields a high-quality subset by eliminating blurry and over-small humans. Unlike existing models that mostly train on full-body humans of simple context (Zhang & Agrawala, 2023), our dataset encompasses a wider spectrum, including various backgrounds and partial human regions such as clothing and limbs.

**2D Human Poses.** 2D human poses (skeleton of joints), which serve as one of the most flexible and easiest obtainable coarse-level condition signals, are widely used in controllable human generation studies (Ju et al., 2023b; Zhu et al., 2023; Yu et al., 2023; Liu et al., 2023; 2022a;b;c). To achieve accurate keypoint annotations, we resort to MMPose (Contributors, 2020) as inference interface and choose ViTPose-H (Xu et al., 2022) as backbone that performs best over several pose estimation benchmarks. In particular, the per-instance bounding box, keypoint coordinates and confidence are labeled, including whole-body skeleton, body skeleton, hand, and facial landmarks.

**Depth and Surface-Normal Maps** are fine-grained structures that reflect the spatial geometry of images (Wu et al., 2022), which are commonly used in conditional generation (Mou et al., 2023). We apply Omnidata (Eftekhar et al., 2021) for monocular depth and normal. The MiDaS (Ranftl et al., 2022) is further annotated following recent depth-to-image pipelines (Rombach et al., 2022).

**Outpaint for Accurate Annotations.** Diffusion models have shown promising results on image inpainting and outpainting, where the appearance and structure of unseen regions can be hallucinated based on the visible parts. Motivated by this, we propose to outpaint each image for a more holistic view given that most off-the-shelf structure estimators are trained on the "complete" image views. Although the outpainted region may be imperfect with artifacts, it can complement a more comprehensive human structure. To this end, we utilize the powerful *SD-Inpaint* to outpaint the surrounding areas of the original canvas. These images are further processed by off-the-shelf estimators, where we only use the labeling within the original image region for more accurate annotations.

**Overall Statistics.** In summary, COYO subset contains $90, 948, 474$ (91M) images and LAION-2B subset contains $248, 396, 109$ (248M) images, which is $18.12\%$ and $20.77\%$ of fullset, respectively. The whole annotation process takes 640 16/32G NVIDIA V100 GPUs for two weeks in parallel.

## 5 EXPERIMENTS

**Experimental Settings.** For the comprehensive evaluation, we divide our comparisons into two settings: **1)** *Quantitative analysis*. All the methods are tested on the same benchmark, using the same prompt with DDIM Scheduler (Song et al., 2020a) for 50 denoising steps to generate the same resolution images of $512 \times 512$. **2)** *Qualitative analysis*. We generate high-resolution $1024 \times 1024$ results for each model with the officially provided best configurations, such as the prompt engineering, noise scheduler, and classifier-free guidance (CFG) scale. Note that we use the RGB output of the first-stage *Latent Structural Diffusion Model* for numerical comparison, while the improved results from the second-stage *Structure-Guided Refiner* are merely utilized for visual comparison.

**Datasets.** We follow common practices in T2I generation (Yu et al., 2022a) and filter out a human subset from MS-COCO 2014 validation (Lin et al., 2014) for zero-shot evaluation. In particular, off-the-shelf human detector and pose estimator are used to obtain $8, 236$ images with clearly-visible humans for evaluation. All the ground truth images are resized and center-cropped to $512 \times 512$. To guarantee fair comparisons, we train first-stage *Latent Structural Diffusion* on *HumanVerse*, which is a subset of public LAION-2B and COYO, to report quantitative metrics. In addition, an internal dataset is adopted to train second-stage *Structure-Guided Refiner* only for visually pleasing results.

Table 1: **Zero-Shot Evaluation on MS-COCO 2014 Validation Human**. We compare our model with recent SOTA general T2I models (Rombach et al., 2022; Podell et al., 2023; DeepFloyd, 2023) and controllable methods (Zhang & Agrawala, 2023; Mou et al., 2023; Ju et al., 2023b). Note that †SDXL generates artistic style in 512, and ‡IF only creates fixed-size images, we first generate $1024 \times 1024$ results, then resize back to $512 \times 512$ for these two methods. We bold the **best** and underline the second results for clarity. Our improvements over the second method are shown in red.

| Methods | Image Quality | | | Alignment | Pose Accuracy | | | |
|---|---|---|---|---|---|---|---|---|
| | FID ↓ | KID$_{\times 1k}$ ↓ | FID$_{CLIP}$ ↓ | CLIP ↑ | AP ↑ | AR ↑ | AP$_{clean}$ ↑ | AR$_{clean}$ ↑ |
| SD 1.5 | 24.26 | 8.69 | 12.93 | 31.72 | - | - | - | - |
| SD 2.0 | 22.98 | 9.45 | 11.41 | 32.13 | - | - | - | - |
| SD 2.1 | 24.63 | 9.52 | 15.01 | 32.11 | - | - | - | - |
| SDXL† | 29.08 | 12.16 | 19.00 | **32.90** | - | - | - | - |
| DeepFloyd-IF‡ | 29.72 | 15.27 | 17.01 | 32.11 | - | - | - | - |
| ControlNet | 27.16 | 10.29 | 15.59 | 31.60 | 20.46 | 30.23 | 25.92 | 38.67 |
| T2I-Adapter | 23.54 | 7.98 | 11.95 | 32.16 | 27.54 | 36.62 | 34.86 | 46.53 |
| HumanSD | 52.49 | 33.96 | 21.11 | 29.48 | 26.71 | 36.85 | 32.84 | 45.87 |
| **HyperHuman** | **17.18** $_{25.2\%\downarrow}$ | **4.11** $_{48.5\%\downarrow}$ | **7.82** $_{31.5\%\downarrow}$ | 32.17 | **30.38** | **37.84** | **38.84** | **48.70** |

**Comparison Methods.** We compare with two categories of open-source SOTA works: **1)** General T2I models, including SD (Rombach et al., 2022) (*SD 1.x & 2.x*), SDXL (Podell et al., 2023), and IF (DeepFloyd, 2023). **2)** Controllable methods with pose condition. Notably, ControlNet (Zhang & Agrawala, 2023) and T2I-Adapter (Mou et al., 2023) can handle multiple structural signals like canny, depth, and normal, where we take their skeleton-conditioned variant for comparison. HumanSD (Ju et al., 2023b) is the most recent work that specializes in pose-guided human generation.

**Implementation Details.** We resize and random-crop the RGB, depth, and normal to the target resolution of each stage. To enforce the model with size and location awareness, the original image height/width and crop coordinates are embedded in a similar way to time embedding (Podell et al., 2023). Our code is developed based on diffusers (von Platen et al., 2022). **1)** For the *Latent Structural Diffusion*, we fine-tune the whole UNet from the pretrained *SD-2.0-base* to **v**-prediction (Salimans & Ho, 2022) in $512 \times 512$ resolution. The DDIMScheduler with improved noise schedule is used for both training and sampling. We train on 128 80G NVIDIA A100 GPUs in a batch size of $2,048$ for one week. **2)** For the *Structure-Guided Refiner*, we choose *SDXL-1.0-base* as the frozen backbone and fine-tune to $\epsilon$-prediction for high-resolution synthesis of $1024 \times 1024$. We train on 256 80G NVIDIA A100 GPUs in a batch size of $2,048$ for one week. The whole two-stage inference process takes 12 seconds on a single 40G NVIDIA A100 GPU. The overall framework is optimized with AdamW (Kingma & Ba, 2015) in $1e-5$ learning rate, and 0.01 weight decay.

## 5.1 MAIN RESULTS

**Evaluation Metrics.** We adopt commonly-used metrics to make comprehensive comparisons from three perspectives: **1)** *Image Quality*. FID, KID, and FID$_{CLIP}$ are used to reflect quality and diversity. **2)** *Text-Image Alignment*, where the CLIP similarity between text and image embeddings is reported. **3)** *Pose Accuracy*. We use the state-of-the-art pose estimator to extract poses from synthetic images and compare with the input (GT) pose conditions. The Average Precision (AP) and Average Recall (AR) are adopted to evaluate the pose alignment. Note that due to the noisy pose estimation of in-the-wild COCO, we also use AP$_{clean}$ and AR$_{clean}$ to only evaluate on the three most salient persons.

**Quantitative Analysis.** We report zero-shot evaluation results in Tab. 1. For all methods, we use the default CFG scale of 7.5, which well balances the quality and diversity with appealing results. Thanks to the structural awareness from expert branches, our proposed **HyperHuman** outperforms previous works by a clear margin, achieving the best results on image quality and pose accuracy metrics and ranks second on CLIP score. Note that SDXL (Podell et al., 2023) uses two text encoders with $3\times$ larger UNet of more cross-attention layers, leading to superior text-image alignment. In spite of this, we still obtain an on-par CLIP score and surpass all the other baselines that have similar text encoder parameters. We also show the FID-CLIP and FID$_{CLIP}$-CLIP curves over multiple CFG scales in Fig. 3, where our model balances well between image quality and text-alignment, especially for the commonly-used CFG scales (*bottom right*). Please see Sec. A.1 for more quantitative results.

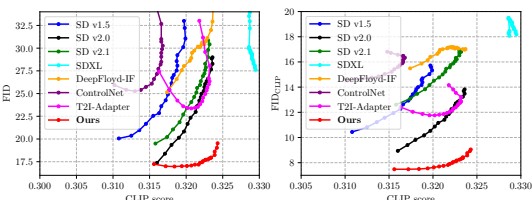

Figure 3: **Evaluation Curves on COCO-Val Human**. We show FID-CLIP (*left*) and FID$_{CLIP}$-CLIP (*right*) curves with CFG scale ranging from 4.0 to 20.0 for all methods.

Table 2: **Ablation Results.** We explore design choices for simultaneous denoising targets, number of expert branch layers, and noise schedules. The image quality and alignment are evaluated.

| Ablation Settings | FID ↓ | FID$_{CLIP}$ ↓ | $\mathcal{L}_2^{\mathbf{d}}$ ↓ | $\mathcal{L}_2^{\mathbf{n}}$ ↓ |
|---|---|---|---|---|
| Denoise RGB | 21.68 | 10.27 | - | - |
| Denoise RGB + Depth | 19.89 | 9.30 | 544.2 | - |
| Denoise RGB + Normal | 19.24 | 9.15 | - | 130.6 |
| Half *DownBlocks* & *UpBlocks* | 22.85 | 11.38 | 508.3 | 124.3 |
| Two *DownBlocks* & *UpBlocks* | 17.94 | 8.85 | 677.4 | 145.9 |
| Default SNR with $\epsilon$-pred | 17.70 | 8.41 | 867.0 | 180.2 |
| Different Timesteps $t$ | 29.36 | 18.29 | 854.8 | 176.1 |
| **HyperHuman (Ours)** | **17.18** | **7.82** | **502.1** | **121.6** |

Table 3: **User Preference Comparisons.** We report the ratio of users prefer our model to baselines.

| Methods | SD 2.1 | SDXL | IF | ControlNet | T2I-Adapter | HumanSD |
|---|---|---|---|---|---|---|
| **HyperHuman** | 89.24% | 60.45% | 82.45% | 92.33% | 98.06% | 99.08% |

**Qualitative Analysis.** Fig. 1 shows results (*top*) and comparisons with baselines (*bottom*). We can generate both photo-realistic images and stylistic rendering, showing better realism, quality, diversity, and controllability. A comprehensive user study is further conducted as shown in Tab. 3, where the users prefer **HyperHuman** to the general and controllable T2I models. Please refer to Appendix A.4, A.15, and A.16 for more user study details, comparisons, and qualitative results.

## 5.2 ABLATION STUDY

In this section, we present the key ablation studies. Except for the image quality metrics, we also use the depth/normal prediction error as a proxy for spatial alignment between the synthesized RGB and structural maps. Specifically, we extract the depth and surface-normal by off-the-shelf estimator as pseudo ground truth. The $\mathcal{L}_2^{\mathbf{d}}$ and $\mathcal{L}_2^{\mathbf{n}}$ denote the $\mathcal{L}_2$-error of depth and normal, respectively.

**Simultaneous Denoise with Expert Branch.** We explore whether latent structural diffusion model helps, and how many layers to replicate in the structural expert branches: **1)** *Denoise RGB*, that only learns to denoise an image. **2)** *Denoise RGB + Depth*, which also predicts depth. **3)** *Denoise RGB + Normal*, **which also predicts surface-normal map**. **4)** *Half DownBlock & UpBlock*. We replicate half of the first *DownBlock* and the last *UpBlock*, which contains one down/up-sample *ResBlock* and one *AttnBlock*. **5)** *Two DownBlocks & UpBlocks*, where we copy the first two *DownBlocks* and the last two *UpBlocks*. The results are shown in Tab. 2 (*top*), which prove that the joint learning of image appearance, spatial relationship, and geometry is beneficial. We also find that while fewer replicate layers give more spatially aligned results, the per-branch parameters are insufficient to capture distributions of each modality. In contrast, excessive replicate layers lead to less feature fusion across different targets, which fails to complement to each other branches.

**Noise Schedules.** The ablation is conducted on two settings: **1)** *Default SNR with $\epsilon$-pred*, where we use the original noise sampler schedules with $\epsilon$-prediction. **2)** *Different Timesteps $t$*. We sample different noise levels ($t_{\mathbf{x}}$, $t_{\mathbf{d}}$, and $t_{\mathbf{n}}$) for each modality. We can see from Tab. 2 (*bottom*) that zero-terminal SNR is important for learning of monotonous structural maps. Besides, different timesteps harm the performance with more sparse perturbation sampling and harder information sharing.

## 6 DISCUSSION

**Conclusion.** In this paper, we propose a novel framework **HyperHuman** to generate in-the-wild human images of high quality. To enforce the joint learning of image appearance, spatial relationship, and geometry in a unified network, we propose *Latent Structural Diffusion Model* that simultaneously denoises the depth and normal along with RGB. Then we devise *Structure-Guided Refiner* to compose the predicted conditions for detailed generation. Extensive experiments demonstrate that our framework yields superior performance, generating realistic humans under diverse scenarios.

**Limitation and Future Work.** As an early attempt in human generation foundation model, our approach creates controllable human of high realism. However, due to the limited performance of existing pose/depth/normal estimators for in-the-wild humans, we find it sometimes fails to generate subtle details like finger and eyes. Besides, the current pipeline still requires body skeleton as input, where deep priors like LLMs can be explored to achieve text-to-pose generation in future work.

## 7 ACKNOWLEDGEMENT

This study is supported by the Ministry of Education, Singapore, under its MOE AcRF Tier 2 (MOE-T2EP20221- 0012) and NTU NAP.

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

# A APPENDIX

In this supplemental document, we provide more details of the following contents: **1)** Additional quantitative results (Sec. A.1). **2)** More implementation details like network architecture, hyper-parameters, and training setups, *etc* (Sec. A.2). **3)** More ablation study results (Sec. A.3). **4)** More user study details (Sec. A.4). **5)** The impact of random seed to our model to show the robustness of our method (Sec. A.5). **6)** Boarder impact and the ethical consideration of this work (Sec. A.6). **7)** Model's robustness on the unseen and challenging pose (Sec. A.7). **8)** Potential optimization for the annotation and training pipeline (Sec. A.8). textbf9) Model's performance on unconditional generation without input poses (Sec. A.9). **10)** Model's performance on the jittered poses and image animation results (Sec. A.10). textbf11) More first-stage generation results (Sec. A.11). **12)** The detailed intuition of updated noise schedule (Sec. A.12). **13)** More details on pose processing and encoding (Sec. A.13). **14)** Reconstruction performance of RGB VAE on other modality-specific inputs (Sec. A.14). **15)** More visual comparison results with recent T2I models (Sec. A.15). **16)** More qualitative results of our model (Sec. A.16). **17)** The asset licenses we use in this work (Sec. A.17).

## A.1 ADDITIONAL QUANTITATIVE RESULTS

**FID-CLIP Curves.** Due to the page limit, we only show tiny-size FID-CLIP and $FID_{CLIP}$-CLIP curves in the main paper and omit the curves of HumanSD (Ju et al., 2023b) due to its too large FID and $FID_{CLIP}$ results for reasonable axis scale. Here, we show a clearer version of FID-CLIP and $FID_{CLIP}$-CLIP curves in Fig. 4. As broadly proven in recent text-to-image studies (Rombach et al., 2022; Nichol et al., 2021; Saharia et al., 2022), the classifier-free guidance (CFG) plays an important role in trading-off image quality and diversity, where the CFG scales around $7.0 - 8.0$ (corresponding to the *bottom-right part* of the curve) are the commonly-used choices in practice. We can see from Fig. 4 that our model can achieve a competitive CLIP Score while maintaining superior image quality results, showing the efficacy of our proposed **HyperHuman** framework.

**Human Preference-Related Metrics.** As shown in recent text-to-image generation evaluation studies, conventional image quality metrics like FID (Heusel et al., 2017), KID (Bińkowski et al., 2018) and text-image alignment CLIP Score (Radford et al., 2021) diverge a lot from the human preference (Kirstain et al., 2023). To this end, we adopt two very recent human preference-related metrics: **1)** PickScore (Kirstain et al., 2023), which is trained on the side-by-side comparisons of two T2I models. **2)** HPS (Human Preference Score) V2 (Wu et al., 2023), which takes the user like/dislike statistics for scoring model training. The evaluation results are reported in Tab. 4, which show that our framework performs better than the baselines. Although the improvement seems to be marginal, we find current human preference-related metrics to be highly biased: The scoring models are mostly trained on the synthetic data with highest resolution of $1024 \times 1024$, which makes them favor unrealistic images of 1024 resolution, as they rarely see real images of higher resolution in score model training. In spite of this, we still achieve superior quantitative and qualitative results on these two metrics and a comprehensive user study, outperforming all the baseline methods.

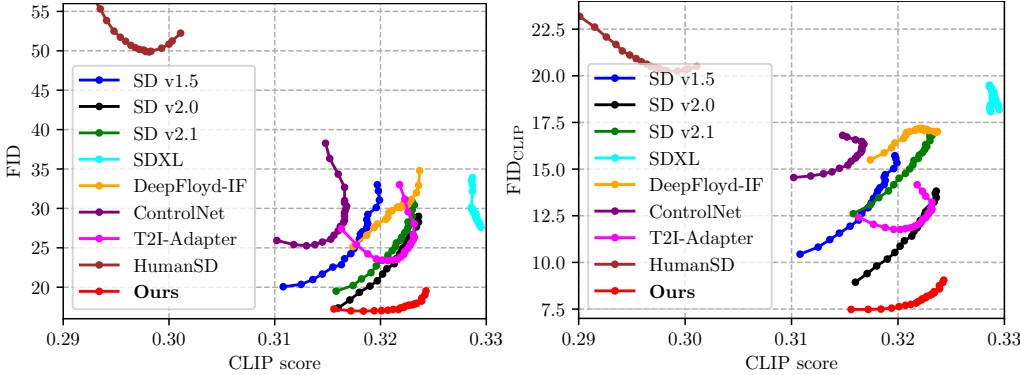

Figure 4: **Clear Evaluation Curves on MS-COCO2014 Validation Human**. We show FID-CLIP (*left*) and $FID_{CLIP}$-CLIP (*right*) curves with CFG scale ranging from 4.0 to 20.0 for all methods.

Table 4: **Quantitative Results on Human Preference-Related Metrics.** We report on two recent metrics PickScore and HPS V2. The first row denotes the ratio of preferring ours to others, where larger than $50\%$ means the superior one. The second row is the human preference score, where the higher the better. It can be seen that our proposed **HyperHuman** achieves the best performance.

| Methods | Ours | SD 2.1 | SDXL | IF | ControlNet | Adapter | HumanSD |
|---------|------|--------|------|-----|-----------|---------|---------|
| PickScore | - | 66.87% | 52.11% | 63.37% | 74.47% | 83.25% | 87.18% |
| HPS V2 | **0.2905** | 0.2772 | 0.2832 | 0.2849 | 0.2783 | 0.2732 | 0.2656 |

**Pose Accuracy Results on Different CFG Scales.** We additionally report the pose accuracy results over different CFG scales. Specifically, we evaluate the conditional human generation methods of ControlNet (Zhang & Agrawala, 2023), T2I-Adapter (Mou et al., 2023), HumanSD (Ju et al., 2023b), and ours on four metrics Average Precision (AP), Average Recall (AR), clean AP ($AP_{clean}$), and clean AR ($AR_{clean}$) as mentioned in Sec. 5.1. We report on CFG scales ranging from 4.0 to 13.0 in Tab. 5, where our method is constantly better in terms of pose accuracy and controllability.

## A.2 MORE IMPLEMENTATION DETAILS

We report implementation details like training hyper-parameters, and model architecture in Tab. 6.

## A.3 MORE ABLATION STUDY RESULTS

We implement additional ablation study experiments on the second stage *Structure-Guided Refiner*. Note that due to the training resource limit and the resolution discrepancy between MS-COCO real images ($512 \times 512$) and high-quality renderings ($1024 \times 1024$), we conduct several toy ablation experiments in the lightweight $512 \times 512$ variant of our model: **1)** *w/o random dropout*, where the all the input conditions are not dropout or masked out during the conditional training stage. **2)** *Only Text*, where not any structural prediction is input to the model and we only use the text prompt as condition. **3)** *Condition on* $\mathbf{p}$, where we only use human pose skeleton $\mathbf{p}$ as input condition to the refiner network. **4)** *Condition on* $\mathbf{d}$ that uses depth map $\mathbf{d}$ as input condition. **5)** *Condition on* $\mathbf{n}$ that uses surface-normal $\mathbf{n}$ as input condition. And their combinations of **6)** *Condition on* $\mathbf{p}$, $\mathbf{d}$; **7)** *Condition on* $\mathbf{p}$, $\mathbf{n}$; **8)** *Condition on* $\mathbf{d}$, $\mathbf{n}$, to verify the impact of each condition and the necessity of using such multi-level hierarchical structural guidance for fine-grained generation. The results are reported in Tab. 7. We can see that the random dropout conditioning scheme is crucial for more robust training with better image quality, especially in the two-stage generation pipeline. Besides, the structural map/guidance contains geometry and spatial relationship information, which are beneficial to image generation of higher quality. Another interesting phenomenon is that only conditioned on surface-normal $\mathbf{n}$ is better than conditioned on both the pose skeleton $\mathbf{p}$ and depth map $\mathbf{d}$, which aligns with our intuition that surface-normal conveys rich structural information that mostly cover coarse-level skeleton and depth map, except for the keypoint location and foreground-background relationship. Overall, we can conclude from ablation results that: **1)** Each condition (*i.e.,* pose skeleton, depth map, and surface-normal) is important for higher-quality and more aligned generation, which validates the necessity of our first-stage *Latent Structural Diffusion Model* to jointly capture them. **2)** The random dropout scheme for robust conditioning can essentially bridge the train-test error accumulation in two-stage pipeline, leading to better image results.

## A.4 MORE USER STUDY DETAILS

The study involves 25 participants and annotates for a total of 8236 images in the zero-shot MS-COCO 2014 validation human subset. They take 2-3 days to complete all the user study task, with a final review to examine the validity of human preference. Specifically, we conduct side-by-side comparisons between our generated results and each baseline model's results. The asking question is **"Considering both the image aesthetics and text-image alignment, which image is better? Prompt: <prompt>."** The labelers are unaware of which image corresponds to which baseline, *i.e.,* the place of two compared images are shuffled to achieve fair comparison without bias.

Table 5: **Additional Pose Accuracy Results for Different CFG Scales**. We evaluate on four pose alignment metrics AP, AR, $AP_{clean}$, and $AR_{clean}$ for the CFG scales ranging from 4.0 to 13.0.

| | CFG 4.0 | | | | CFG 5.0 | | | |
|---|---|---|---|---|---|---|---|---|
| Methods | AP ↑ | AR ↑ | $AP_{clean}$ ↑ | $AR_{clean}$ ↑ | AP ↑ | AR ↑ | $AP_{clean}$ ↑ | $AR_{clean}$ ↑ |
| ControlNet | 20.37 | 29.54 | 25.98 | 37.96 | 20.42 | 29.94 | 26.09 | 38.31 |
| T2I-Adapter | 28.18 | 36.71 | 35.68 | 46.77 | 27.90 | 36.76 | 35.31 | 46.78 |
| HumanSD | 26.05 | 35.89 | 32.27 | 44.90 | 26.51 | 36.44 | 32.84 | 45.48 |
| **HyperHuman** | **30.45** | **37.87** | **38.88** | **48.75** | **30.57** | **37.96** | **39.01** | **48.84** |

| | CFG 6.0 | | | | CFG 7.0 | | | |
|---|---|---|---|---|---|---|---|---|
| Methods | AP ↑ | AR ↑ | $AP_{clean}$ ↑ | $AR_{clean}$ ↑ | AP ↑ | AR ↑ | $AP_{clean}$ ↑ | $AR_{clean}$ ↑ |
| ControlNet | 20.54 | 30.16 | 26.09 | 38.64 | 20.44 | 30.29 | 26.01 | 38.79 |
| T2I-Adapter | 27.90 | 36.77 | 35.37 | 46.80 | 27.66 | 36.62 | 35.00 | 46.55 |
| HumanSD | 26.79 | 36.79 | 33.10 | 45.91 | 26.73 | 36.84 | 32.94 | 45.80 |
| **HyperHuman** | **30.44** | **37.92** | **38.91** | **48.77** | **30.49** | **37.90** | **38.82** | **48.72** |

| | CFG 8.0 | | | | CFG 9.0 | | | |
|---|---|---|---|---|---|---|---|---|
| Methods | AP ↑ | AR ↑ | $AP_{clean}$ ↑ | $AR_{clean}$ ↑ | AP ↑ | AR ↑ | $AP_{clean}$ ↑ | $AR_{clean}$ ↑ |
| ControlNet | 20.54 | 30.28 | 26.06 | 38.74 | 20.35 | 30.11 | 25.80 | 38.43 |
| T2I-Adapter | 27.46 | 36.50 | 34.80 | 46.39 | 27.10 | 36.32 | 34.14 | 46.04 |
| HumanSD | 26.76 | 36.86 | 32.96 | 45.88 | 26.67 | 36.91 | 32.74 | 45.93 |
| **HyperHuman** | **30.23** | **37.80** | **38.72** | **48.59** | **29.93** | **37.67** | **38.30** | **48.45** |

| | CFG 10.0 | | | | CFG 11.0 | | | |
|---|---|---|---|---|---|---|---|---|
| Methods | AP ↑ | AR ↑ | $AP_{clean}$ ↑ | $AR_{clean}$ ↑ | AP ↑ | AR ↑ | $AP_{clean}$ ↑ | $AR_{clean}$ ↑ |
| ControlNet | 20.10 | 30.08 | 25.50 | 38.29 | 19.81 | 29.93 | 25.23 | 38.23 |
| T2I-Adapter | 26.89 | 36.19 | 33.83 | 45.83 | 26.65 | 36.10 | 33.51 | 45.67 |
| HumanSD | 26.67 | 36.86 | 32.80 | 46.00 | 26.53 | 36.74 | 32.63 | 45.85 |
| **HyperHuman** | **29.75** | **37.60** | **38.20** | **48.38** | **29.58** | **37.31** | **37.88** | **48.07** |

| | CFG 12.0 | | | | CFG 13.0 | | | |
|---|---|---|---|---|---|---|---|---|
| Methods | AP ↑ | AR ↑ | $AP_{clean}$ ↑ | $AR_{clean}$ ↑ | AP ↑ | AR ↑ | $AP_{clean}$ ↑ | $AR_{clean}$ ↑ |
| ControlNet | 19.57 | 29.84 | 25.02 | 38.15 | 19.52 | 29.74 | 24.93 | 38.08 |
| T2I-Adapter | 26.49 | 35.95 | 33.39 | 45.52 | 26.41 | 35.90 | 33.22 | 45.44 |
| HumanSD | 26.46 | 36.71 | 32.53 | 45.82 | 26.26 | 36.65 | 32.39 | 45.70 |
| **HyperHuman** | **29.40** | **37.18** | **37.75** | **47.90** | **29.29** | **37.11** | **37.64** | **47.87** |

Table 6: **Training Hyper-parameters and Network Architecture in HyperHuman**.

| | Latent Structural Diffusion | Structure-Guided Refiner |
|---|---|---|
| Activation Function | SiLU | SiLU |
| Additional Embed Type | Time | Text + Time |
| # of Heads in Additional Embed | 64 | 64 |
| Additional Time Embed Dimension | 256 | 256 |
| Attention Head Dimension | [5, 10, 20, 20] | [5, 10, 20] |
| Block Out Channels | [320, 640, 1280, 1280] | [320, 640, 1280] |
| Cross-Attention Dimension | 1024 | 2048 |
| Down Block Types | ["CrossAttn"×3,"ResBlock"×1] | ["ResBlock"×1,"CrossAttn"×2] |
| Input Channel | 8 | 4 |
| # of Input Head | 3 | 3 |
| Condition Embedder Channels | - | [16, 32, 96, 256] |
| Transformer Layers per Block | [1, 1, 1, 1] | [1, 2, 10] |
| Layers per Block | [2, 2, 2, 2] | [2, 2, 2] |
| Input Class Embedding Dimension | — | 2816 |
| Sampler Training Step $T$ | 1000 | 1000 |
| Learning Rate | $1e-5$ | $1e-5$ |
| Weight Decay | 0.01 | 0.01 |
| Warmup Steps | 0 | 0 |
| AdamW Betas | (0.9, 0.999) | (0.9, 0.999) |
| Batch Size | 2048 | 2048 |
| Condition Dropout | 15% | 50% |
| Text Encoder | OpenCLIP ViT-H (Radford et al., 2021) | CLIP ViT-L & OpenCLIP ViT-bigG (Radford et al., 2021) |
| Pretrained Model | SD-2.0-base (Rombach et al., 2022) | SDXL-1.0-base (Podell et al., 2023) |

Table 7: **Additional Ablation Results for Structure-Guided Refiner**. Due to the resource limit and resolution discrepancy, we experiment on $512 \times 512$ resolution to illustrate our design's efficacy.

| Ablation Settings | FID ↓ | KID$_{\times 1k}$ ↓ | FID$_{\text{CLIP}}$ ↓ | CLIP ↑ |
|---|---|---|---|---|
| w/o random dropout | 25.69 | 11.84 | 13.48 | 31.83 |
| Only Text | 23.99 | 10.42 | 13.22 | **32.23** |
| Condition on **p** | 20.97 | 7.51 | 12.86 | 31.95 |
| Condition on **d** | 14.97 | 3.75 | 9.88 | 31.74 |
| Condition on **n** | 12.67 | 2.61 | 7.09 | 31.59 |
| Condition on **p**, **d** | 14.98 | 3.78 | 9.47 | 31.74 |
| Condition on **p**, **n** | 12.65 | 2.66 | 6.93 | 31.63 |
| Condition on **d**, **n** | 12.42 | 2.59 | 6.89 | 31.57 |
| **Ours w/ Refiner** | **12.38** | **2.55** | **6.76** | **32.23** |

We note that all the labelers are well-trained for such text-to-image generation comparison tasks, who have passed the examination on a test set and have experience in this kind of comparisons for over 50 times. Below, we include the user study rating details for our method *vs.* baseline models. Each labeler can click on four options: **a)** *The left image is better*, in this case the corresponding model will get $+1$ grade. **b)** *The right image is better*. **c)** *NSFW*, which means the prompt/image contain NSFW contents, in this case both models will get 0 grade. **d)** *Hard Case*, where the labelers find it hard to tell which one's image quality is better, in this case both models will get $+0.5$ grade. The detailed comparison statistics are shown in Table 8, where we report the grades of **HyperHuman** *vs.* baseline methods. It can be clearly seen that our proposed framework is superior than all the existing models, with better image quality, realism, aesthetics, and text-image alignment.

Table 8: **Detailed Comparison Statistics in User Study.** We conduct a comprehensive user study on zero-shot MS-COCO 2014 validation human subset with well-trained participants.

| Methods | SD 2.1 | SDXL | IF |
|---|---|---|---|
| **HyperHuman** | **7350** *vs.* 886 | **4978.5** *vs.* 3257.5 | **6787.5** *vs.* 1444.5 |

| Methods | ControlNet | T2I-Adapter | HumanSD |
|---|---|---|---|
| **HyperHuman** | **7604** *vs.* 632 | **8076** *vs.* 160 | **8160** *vs.* 76 |

A.5    IMPACT OF RANDOM SEED AND MODEL ROBUSTNESS

To further validate our model's robustness to the impact of random seed, we inference with the same input conditions (*i.e.,* text prompt and pose skeleton) and use different random seeds for generation. The results are shown in Fig. 5, which suggest that our proposed framework is robust to generate high-quality and text-aligned human images over multiple arbitrary random seeds.

A bearded, bald man wears a multicolored tie.

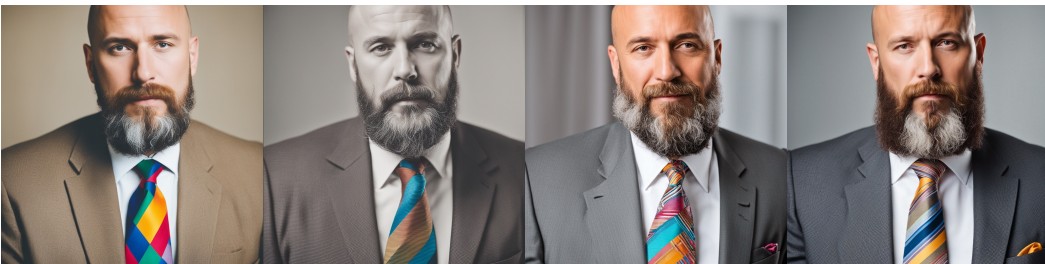

A smiling man in a skiing outfit holds his skis and poles

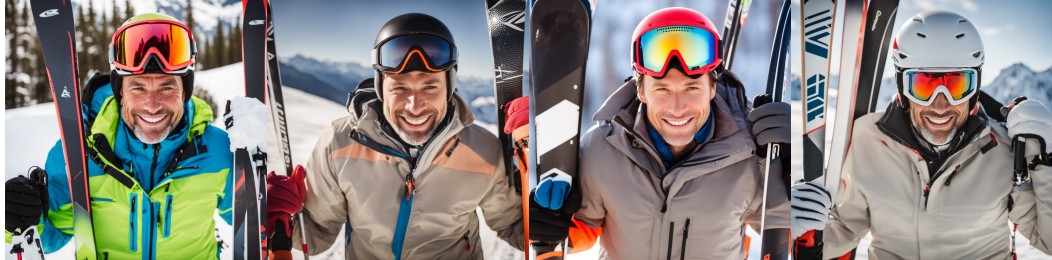

A handsome man holds a glass of white wine and looks at the camera.

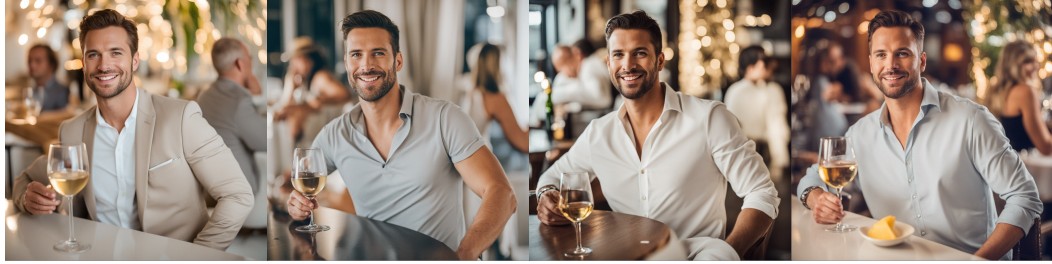

Figure 5: **Impact of Random Seed and Model Robustness.** We use the same input text prompt and pose skeleton with different random seeds to generate multiple results. The results suggest that our proposed framework is robust to generate high-quality and text-aligned human images.

A.6    BOARDER IMPACT AND ETHICAL CONSIDERATION

Generating realistic humans conditioned on text benefits a wide range of applications. It enriches creative domains such as art, design, and entertainment by enabling the creation of highly realistic and emotionally resonant visuals. Besides, it streamlines design processes, reducing time and resources needed for tasks like content production. However, it could be misused for malicious purposes like deepfake or forgery generation. We believe that the proper use of this technique will enhance the machine learning research and digital entertainment. We also advocate all the generated images should be labeled as "synthetic" to avoid negative social impacts.

A.7    MODEL ROBUSTNESS ON UNSEEN AND CHALLENGING POSE

In this section, we show the robustness of **HyperHuman** to generalize to unseen or challenging poses. Specifically, we choose an acrobatic-related image from the Human-Art dataset (Ju et al., 2023a), which is a highly challenging and rare pose unseen from the common human-centric images.

The results are shown in Fig. 6. In the visualized results, **(a)** is the ground-truth image from the Human-Art dataset; **(b)** is the associated pose skeleton, which is challenging and unseen; **(c)**, **(d)**, **(e)**, and **(f)** are four generated images from our proposed framework. It can be seen that **HyperHuman** is robust to unseen poses, even for the rare acrobatic case.

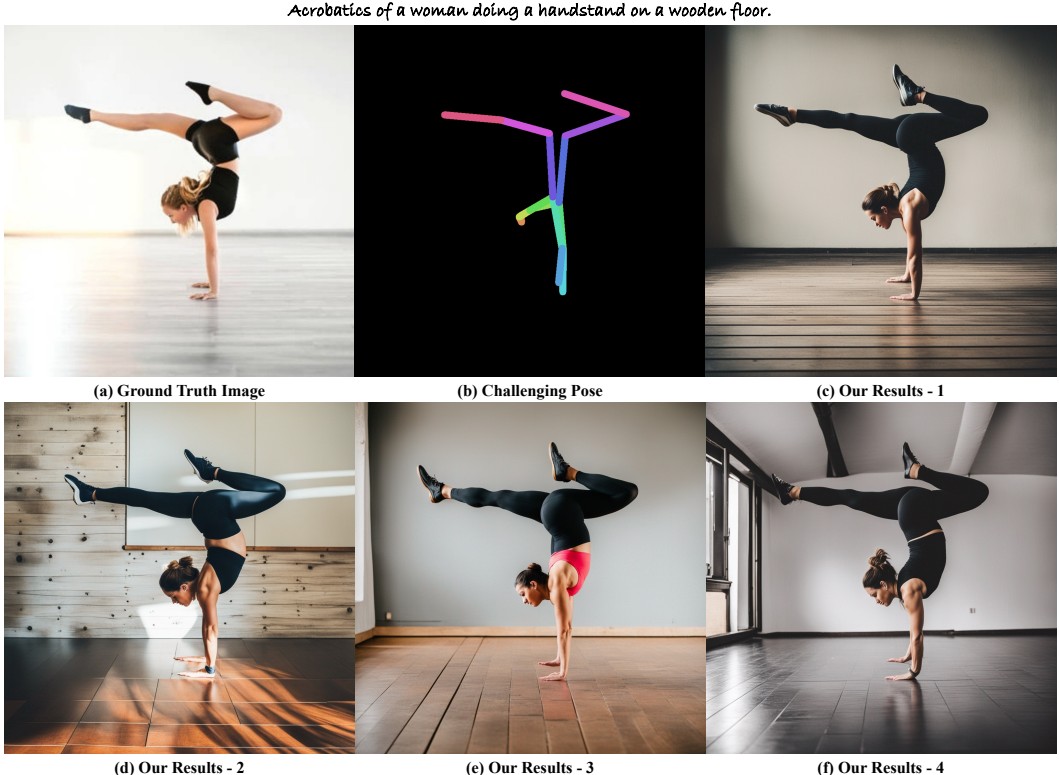

Figure 6: **Model Robustness on Unseen and Challenging Pose.** We show multiple high-quality generation results on the unseen acrobatic pose, which shows the robustness of our method.

## A.8  POTENTIAL OPTIMIZATION FOR ANNOTATION AND TRAINING

From the perspective of optimizing training: **1)** We can change our models into a smaller diffusion backbone to save the training and memory cost, *e.g.*, Small SD and Tiny SD (Kim et al., 2023), which achieve on-par performance with Stable Diffusion, but lighter and faster in training and inference. **2)** We can leverage some efficient parameter finetuning techniques like LoRA (Hu et al., 2021) and Adapter (Houlsby et al., 2019) to finetune the shared backbone with fewer parameters. **3)** We can adopt some common engineering tricks to reduce memory consumption, *e.g.*, gradient checkpointing, gradient accumulation with smaller batch size, deepspeed model parallelism, lower floating point precision like fp16, efficient xformers, *etc*.

From the perspective of optimizing annotation: **1)** Our efficient architecture design (only add lightweight branches) can actually produce reasonable results with smaller dataset scale and fewer training iterations, capturing the joint distribution of RGB, depth, and surface-normal. Before the large-scale training, we first verify method effectiveness on a small-scale $1M$ subset, which is less than 3% of the HumanVerse fullset scale. In spite of this, we can still obtain good results with only 8 40GB A100 within one day, generating spatially aligned results for each modality. A generation sample is shown in Fig. 7, where **(a)** is the conditioning pose skeleton; **(b)**, **(c)**, and **(d)** are the simultaneously denoised depth map, surface-normal map, and RGB images. Note that since this is an early-stage experiment, the pose conditioning and visualization are little bit different from the final version we have used. In spite of this, we manage to achieve simultaneous denoising of multiple modalities with a much smaller dataset scale. **2)** The annotation overhead mostly comes from the diffusion-based image outpainting process (Sec. 4), while the cost for depth and normal estimation is

relatively low. Though facilitating more accurate pose annotations, it is not a mandatory step. Moreover, in the final evaluation process, we use the raw human pose without the help of outpainting, but can still achieve superior performance.

Figure 7: **An Early-Stage Generation Sample on Small-Scale Dataset.** We show a generation sample on a small-scale $1M$ subset, which is less than $3\%$ of the HumanVerse fullset scale. Note that since this is an early-stage experiment, the pose conditioning and visualization are little bit different from the final version we have used.

## A.9 MODEL PERFORMANCE WITHOUT INPUT POSE

In this section, we show the unconditional generation results of our model, where no pose input is taken. The generated images are shown in Fig. 8. All the text prompts are from the zero-shot MS-COCO 2014 Human Validation dataset, which is unseen during the model training process. Thanks to our framework design of robust conditioning scheme, the model is trained to predict reasonable denoising results, even when the conditions are dropout or masked. Therefore, we manage to create realistic human images with superior performance even without the pose skeleton as input.

## A.10 MODEL PERFORMANCE ON JITTERED POSE AND IMAGE ANIMATION

We show additional results on the jittered human poses in Fig. 9. Specifically, we first condition on the original pose skeleton **(a)** and obtain the generated image **(b)** based on text prompt "A woman standing near a lake with a snow capped mountain behind". Then we gradually add Gaussian noise to all the joints, from the sigma scale of 2.5 to 12.5. It can be seen that **HyperHuman** could produce pleasant results under Gaussian noises to all joints, creating highly pose-aligned images.

To further verify if we can animate a certain image by gradually changing the input pose, we fix the random seed, the initial starting noise $\mathbf{x}_T$, and text prompt. The sequential generation results are shown in Fig. 10. Note that we fix the text prompt of "A woman standing near a lake with a snow capped mountain behind". The input skeleton are shifted towards the right side, each step by 10 pixels. Even though we maintain other conditions fixed, we can still see background and appearance changes. We regard this as a promising research problem and will explore it in future work.

## A.11 MORE FIRST-STAGE GENERATION RESULTS

We show more first-stage *Latent Structural Diffusion Model* generation results in Fig. 11, where the spatially aligned RGB images, depth maps, and surface-normal maps are simultaneously denoised and generated. Though not as high-quality as the final output from the second-stage pipeline, it can still generate plausible humans with coherent structures.

## A.12 DETAILED INTUITION OF UPDATED NOISE SCHEDULE

First, it is hard to finetune the Stable Diffusion to generate pure-color images. As shown in the paper (Lin et al., 2023), we can not even overfit to a single solid-black image with the text prompt of "Solid black background". The main reason is that common diffusion noise schedules are flawed, which corrupts image incompletely when sampling $t = T$ at the training phase: $\mathbf{x}_T = \alpha_T \cdot \mathbf{x}_0 + \sigma_T \cdot \epsilon$,

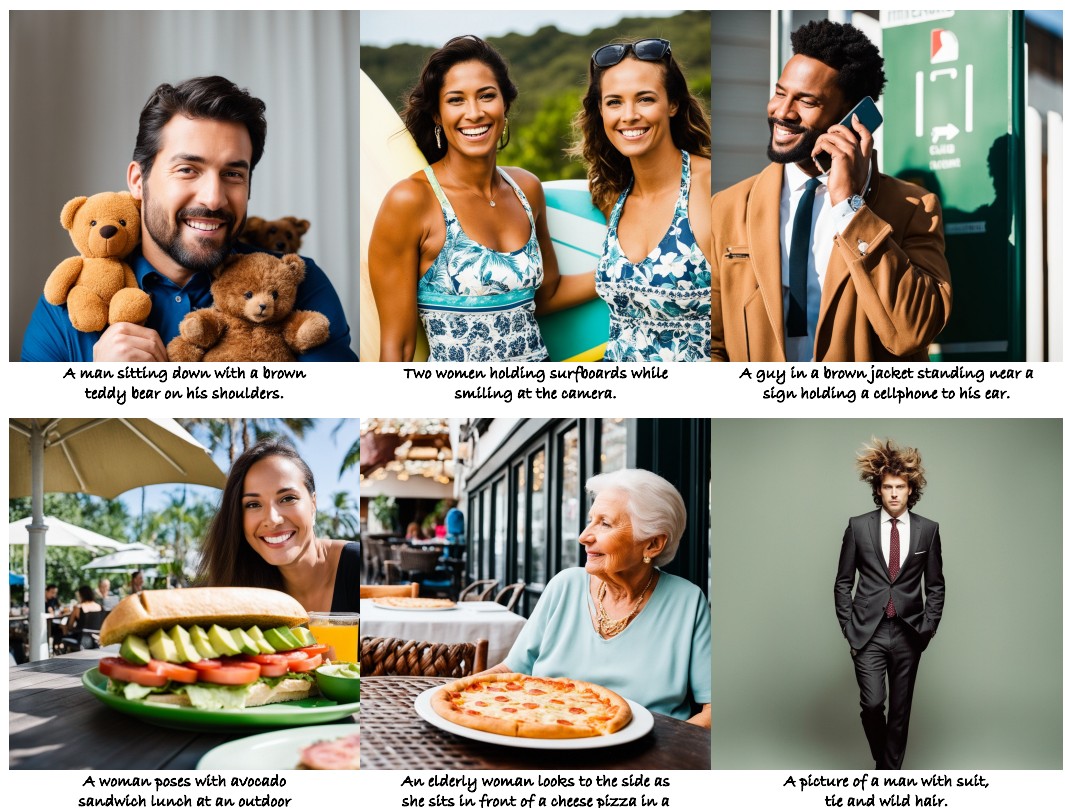

Figure 8: **Unconditional Generation Results without Input Pose.** All the text prompts are from the zero-shot MS-COCO 2014 Human Validation dataset.

but $\alpha_T \neq 0, \sigma_T \neq 1$. Due to this reason, a small amount of signal is still included, which leaks the lowest frequency information such as the overall mean of each channel. In contrast, at the inference stage, the sampling starts from a pure Gaussian noise, which has a zero mean. Such train-test gap hinders SD from generating pure-color images.

Second, similar to pure color images, the depth and surface-normal maps are visualized based on certain scheme, where its color and patterns are highly constrained. For example, the depth map is grey-scale image without colorful textures, and current estimators tend to infer similar depth values for each local patch. Therefore, the low frequency information of per-channel mean and standard deviation could be misused by network as shortcut for denoising, which harms the joint learning of multiple modalities (RGB, depth, and surface-normal). Motivated by this, we propose to enforce the zero-terminal SNR ($\mathbf{x}_T = 0.0 \cdot \mathbf{x}_0 + 1.0 \cdot \epsilon$, that is, $\alpha_T = 0, \sigma_T = 1$) to fully eliminate low-frequency information at the training stage, so that we manage generate both RGB images and structural maps of high quality at the inference stage.

## A.13  MORE DETAILS ON POSE PROCESSING AND ENCODING

The encoder used for pose is the pretrained VAE encoder of Stable Diffusion, which is the same as the encoder used for RGB, depth, and surface-normal maps. Before pose encoding, we visualize the body keypoints on a black canvas to form a skeleton map, similar to previous controllable methods (Zhang & Agrawala, 2023; Mou et al., 2023; Ju et al., 2023b) with pose condition. Specifically, we use exactly the same pose drawing method as HumanSD (Ju et al., 2023b) and T2I-Adapter (Mou et al., 2023) to ensure fairness.

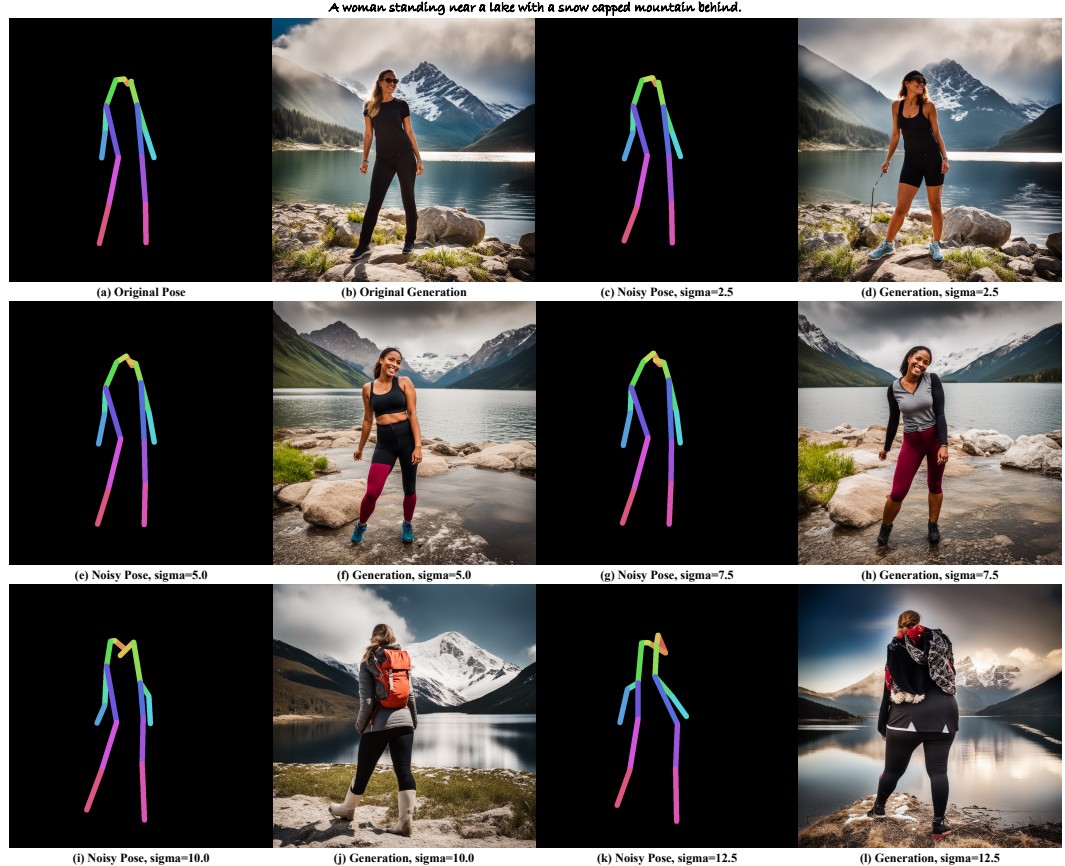

Figure 9: **Generation Results under the Jittered Poses.** We use the text prompt "A woman standing near a lake with a snow capped mountain behind" and gradually add Gaussian noise to all the joints, from the sigma scale of 2.5 to 12.5.

### A.14 VAE RECONSTRUCTION PERFORMANCE ON MODALITY-SPECIFIC INPUT

We use an improved auto-encoder of the pretrained Stable Diffusion "sd-vae-ft-mse"[1] as VAE to encode inputs from all the modalities, including RGB, depth, surface-normal, and body skeleton maps. To further validate that RGB VAE can be directly used for other structural maps, we extensively evaluate the reconstruction metrics of all the involved structural maps on $100k$ samples. The results are reported in Tab. 9, which show that the pre-trained RGB VAE is robust enough to handle different modality images, including the structural maps we use in this work. Besides, we additionally show some visualized reconstruction samples in Fig. 12, where in each group, the first row is the input structural maps, and the second row is the reconstructed structural maps from the pretrained RGB VAE. Therefore, both the quantitative metrics and visual results show that the pretrained RGB VAE is robust enough to faithfully reconstruct structural maps.

Table 9: **RGB VAE Reconstruction Performance**. We evaluate the reconstruction performance of the pretrained RGB VAE on the depth and surface-normal maps.

| Modality | rFID ↓ | PSNR ↑ | SSIM ↑ | PSIM ↓ |
|---|---|---|---|---|
| Body Skeleton | 0.49 | 39.24 | 0.96 | 0.188 |
| MiDaS Depth | 0.19 | 47.08 | 0.99 | 0.004 |
| Surface-Normal | 0.24 | 40.11 | 0.97 | 0.010 |

---

[1]https://huggingface.co/stabilityai/sd-vae-ft-mse

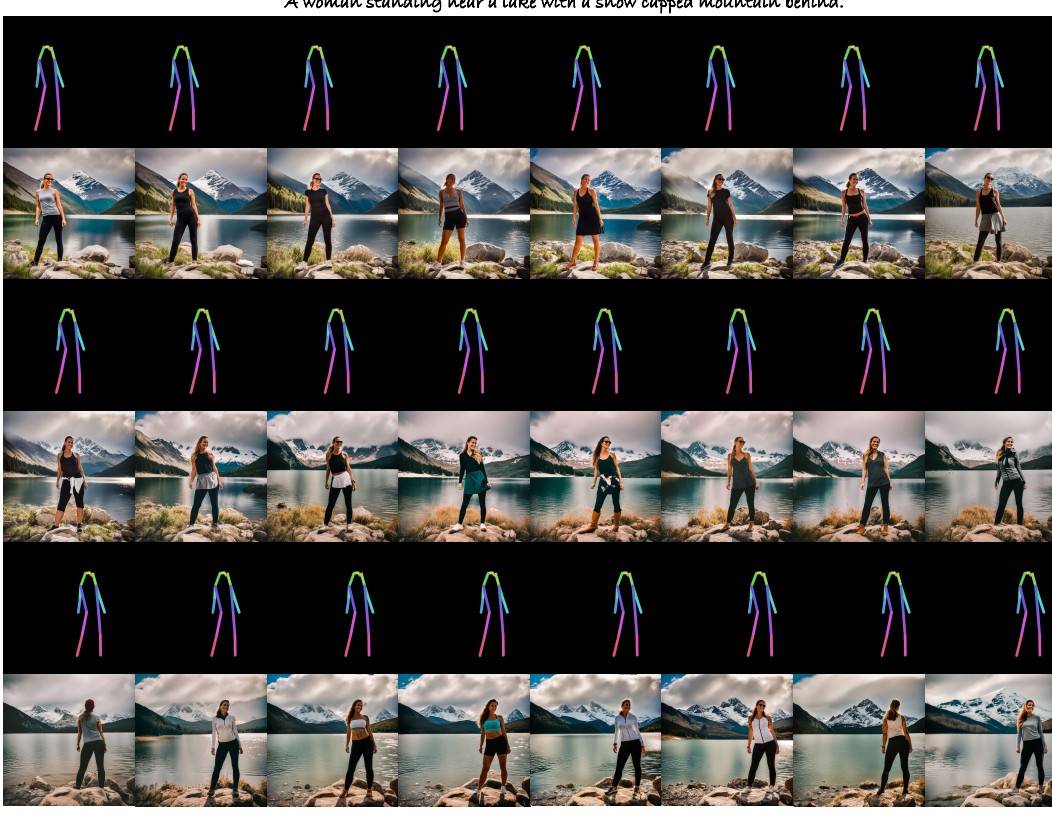

Figure 10: **Animation Results.** We gradually shift skeleton to right side, each step by 10 pixels.

## A.15    MORE COMPARISON RESULTS

We additionally compare our proposed **HyperHuman** with recent open-source general text-to-image models and controllable human generation baselines, including ControlNet (Zhang & Agrawala, 2023), T2I-Adapter (Mou et al., 2023), HumanSD (Ju et al., 2023b), *SD v2.1* (Rombach et al., 2022), DeepFloyd-IF (DeepFloyd, 2023), *SDXL 1.0* w/ refiner (Podell et al., 2023). Besides, we also compare with the concurrently released T2I-Adapter+SDXL[2]. We use the officially-released models to generate high-resolution images of $1024 \times 1024$ for all methods. The results are shown in Fig. 13, 14, 15, and 16, which demonstrates that we can generate humans of high realism.

## A.16    ADDITIONAL QUALITATIVE RESULTS

We further inference on the challenging zero-shot MS-COCO 2014 validation human subset prompts and show additional qualitative results in Fig. 17, 18, and 19. All the images are in high resolution of $1024 \times 1024$. It can be seen that our proposed **HyperHuman** framework manages to synthesize realistic human images of various layouts under diverse scenarios, *e.g.,* different age groups of baby, child, young people, middle-aged people, and old persons; different contexts of canteen, in-the-wild roads, snowy mountains, and streetview, *etc*. Please kindly zoom in for the best viewing.

---

[2] https://huggingface.co/Adapter/t2iadapter

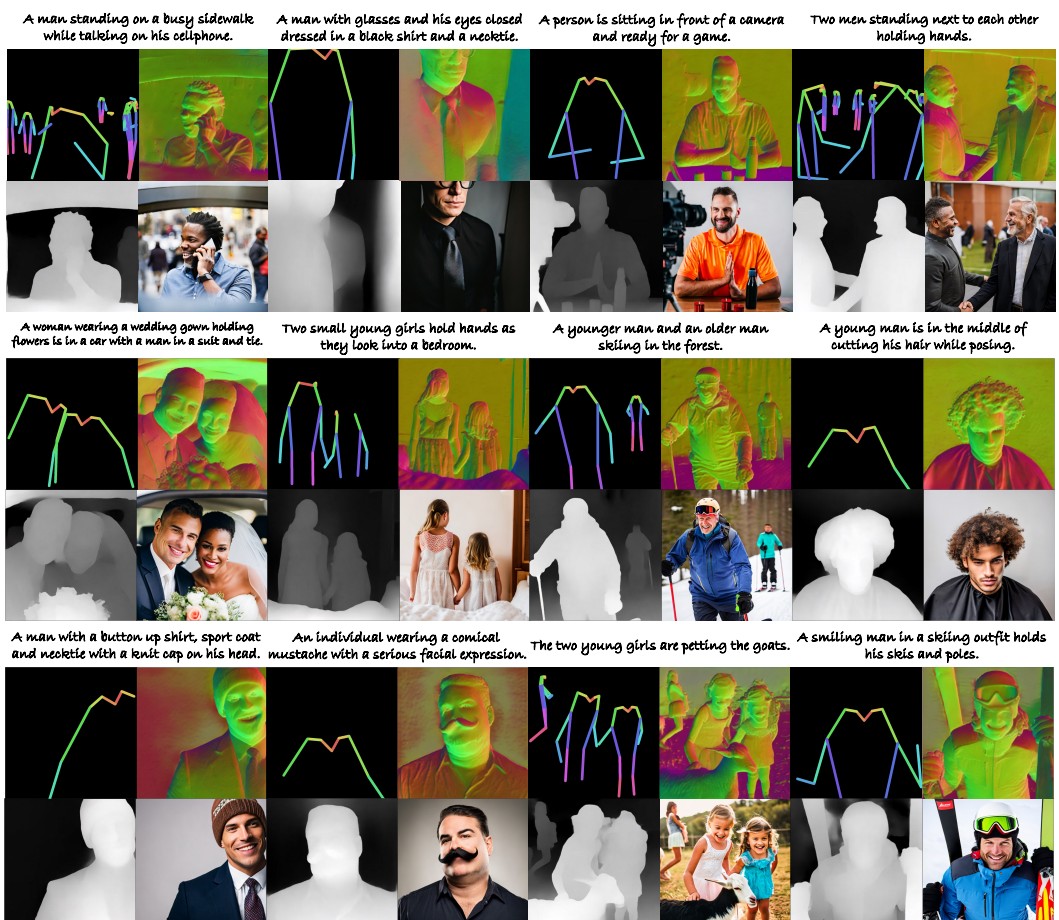

Figure 11: **First-Stage Results.** We show the jointly denoised RGB, depth, and normal maps.

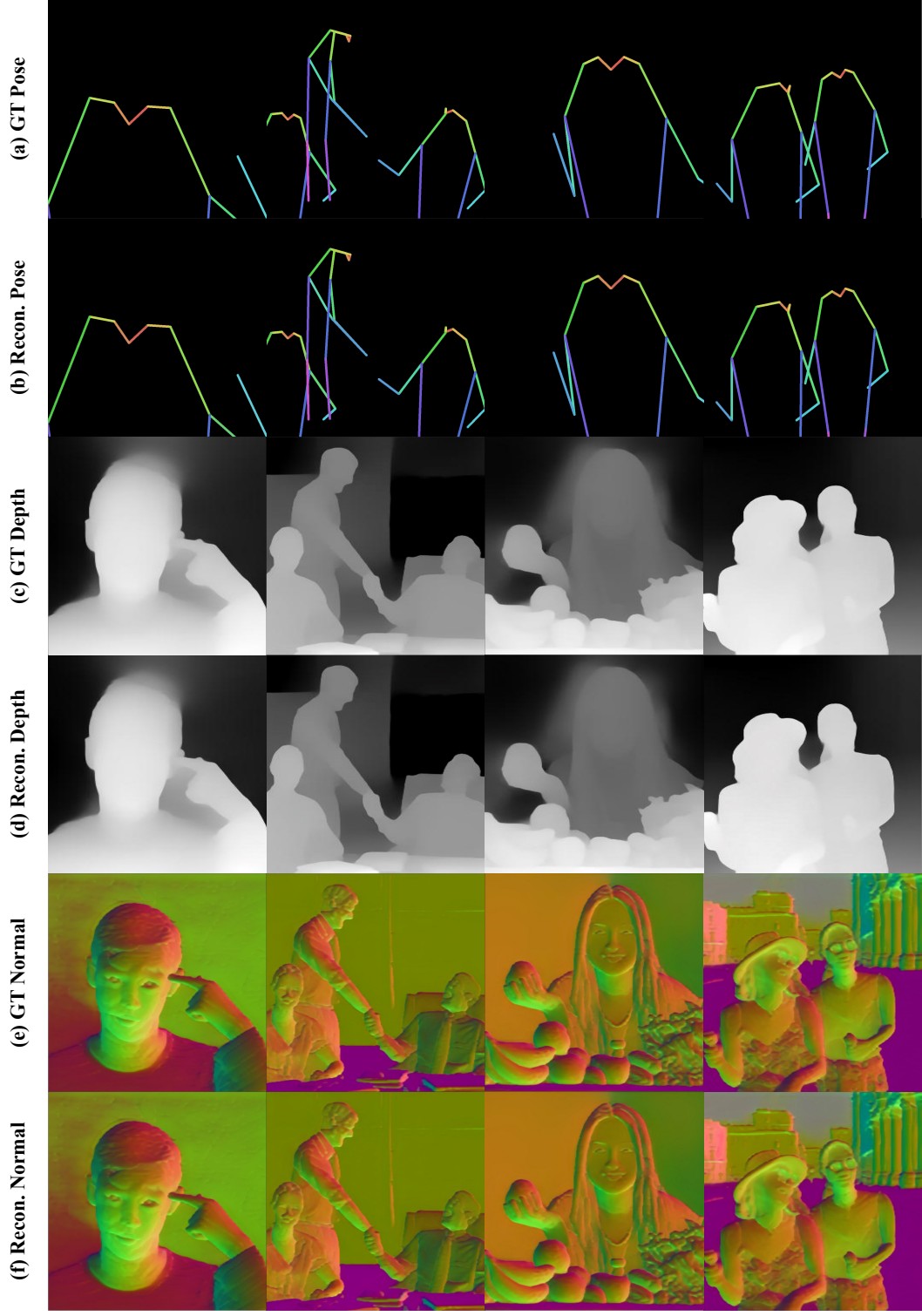

Figure 12: **RGB VAE Reconstruction Results.** We show the visualized reconstruction results on depth and surface-normal maps.

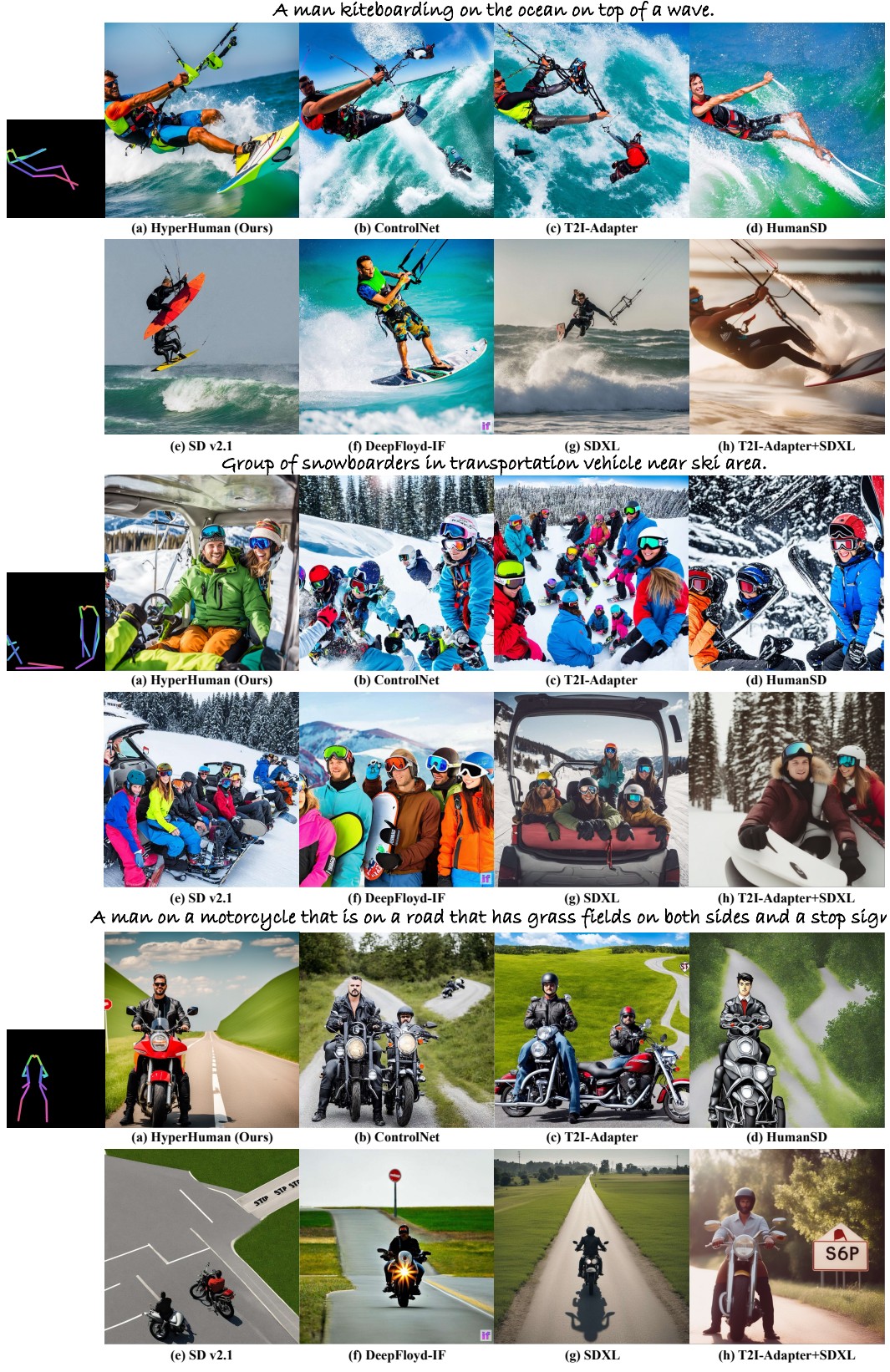

Figure 13: **Additional Comparison Results.**

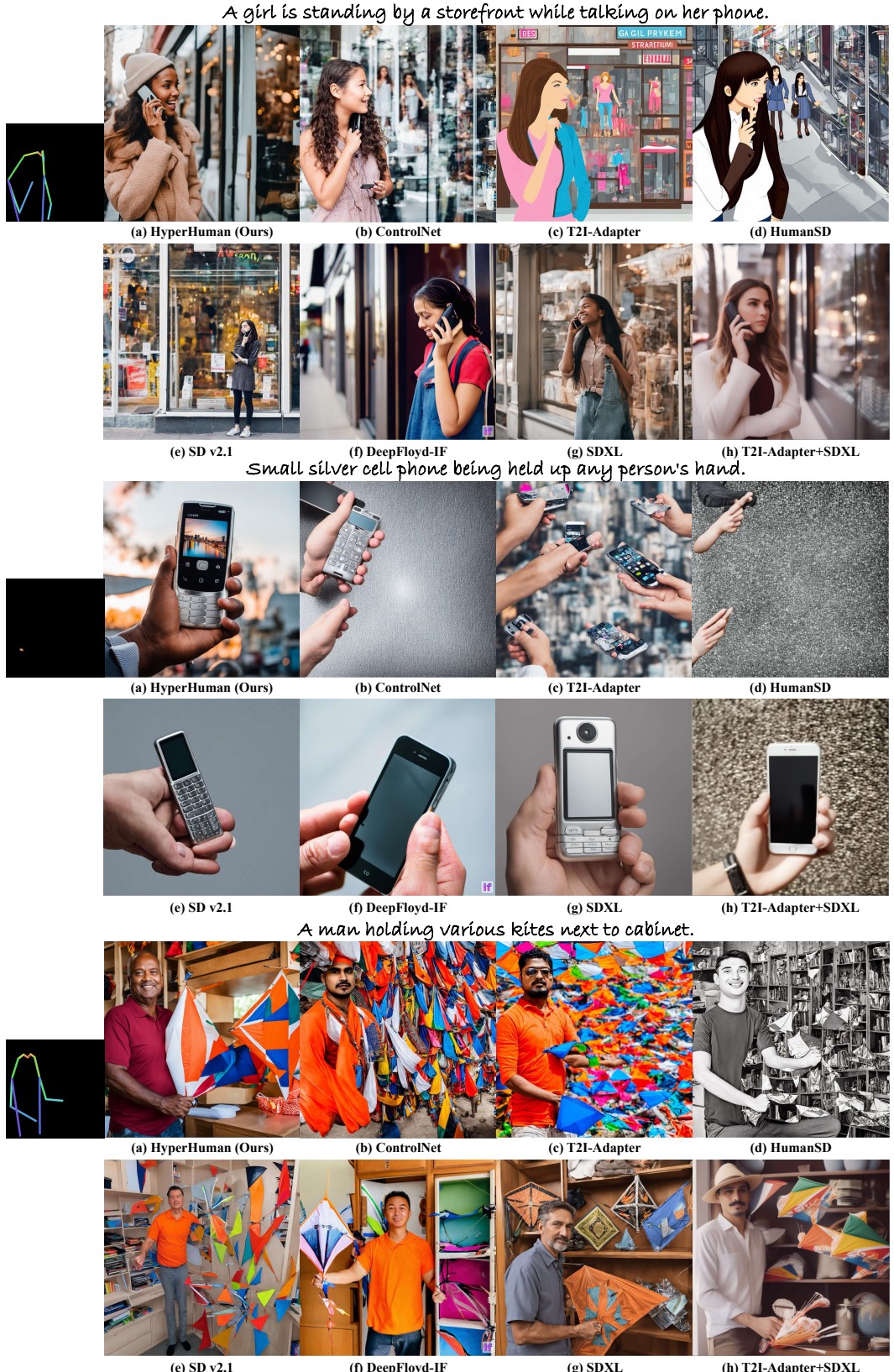

Figure 14: **Additional Comparison Results.**

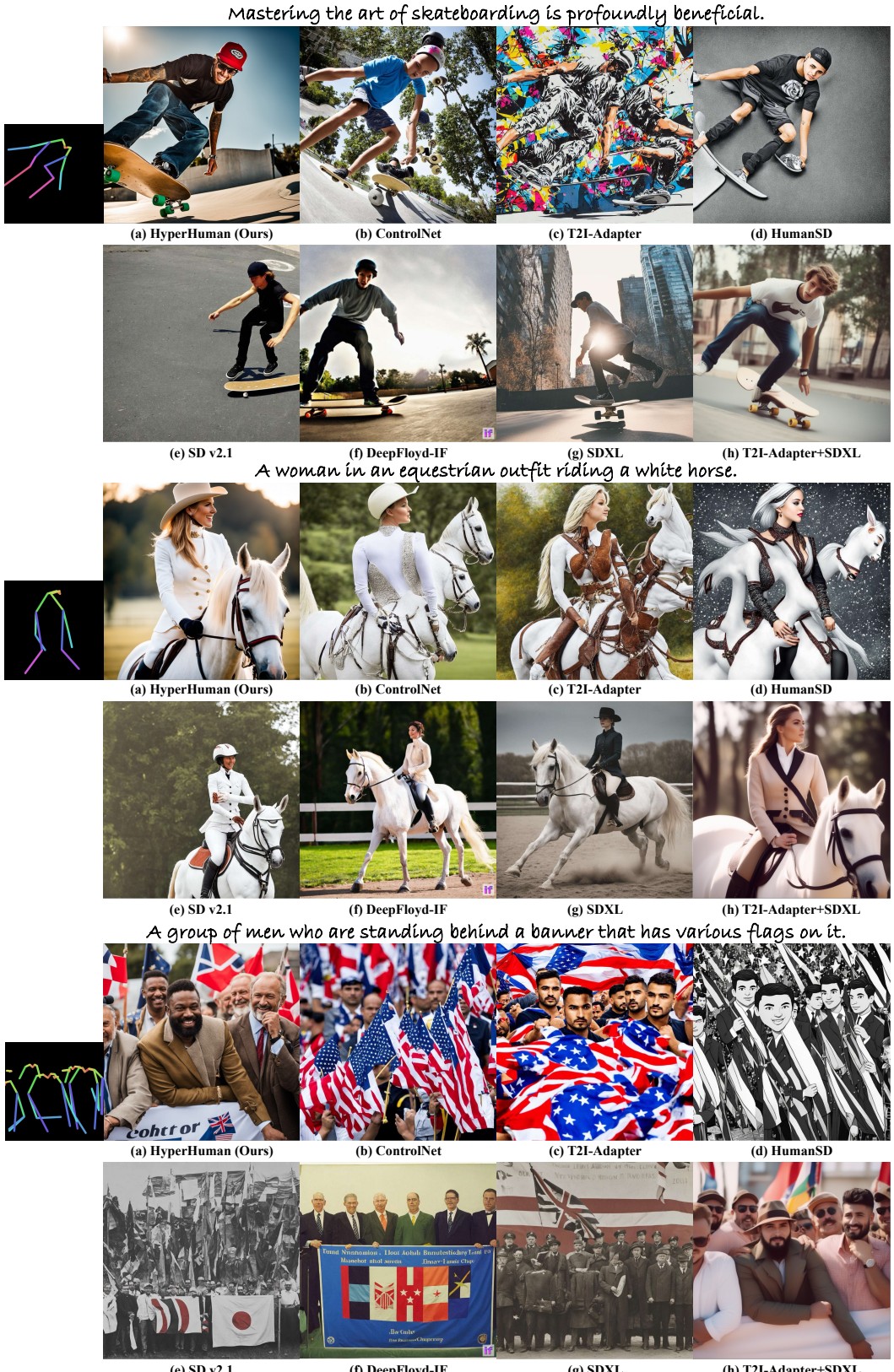

Figure 15: **Additional Comparison Results.**

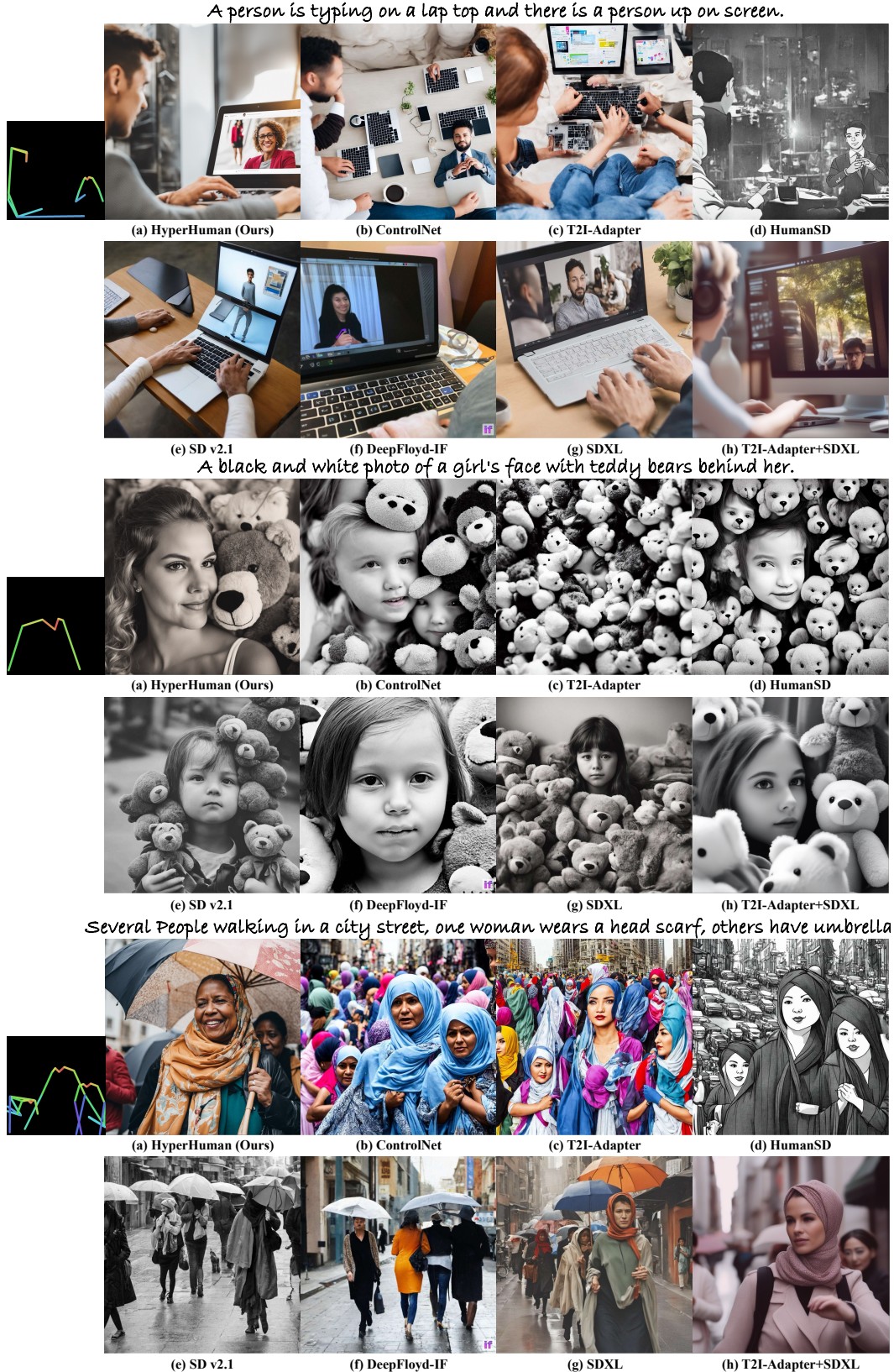

Figure 16: **Additional Comparison Results.**

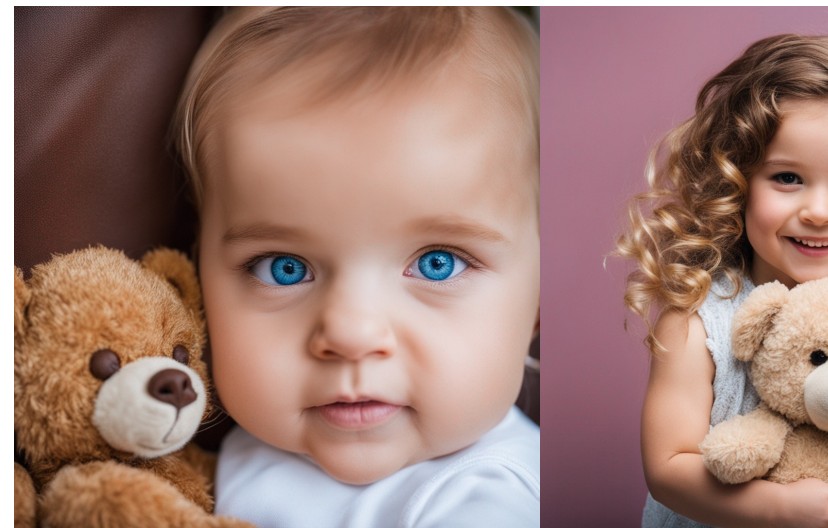

A baby girl with beautiful blue eyes
standing next to a brown teddy bear.

A little girl with wavy hair
and smile holding a teddy bear.

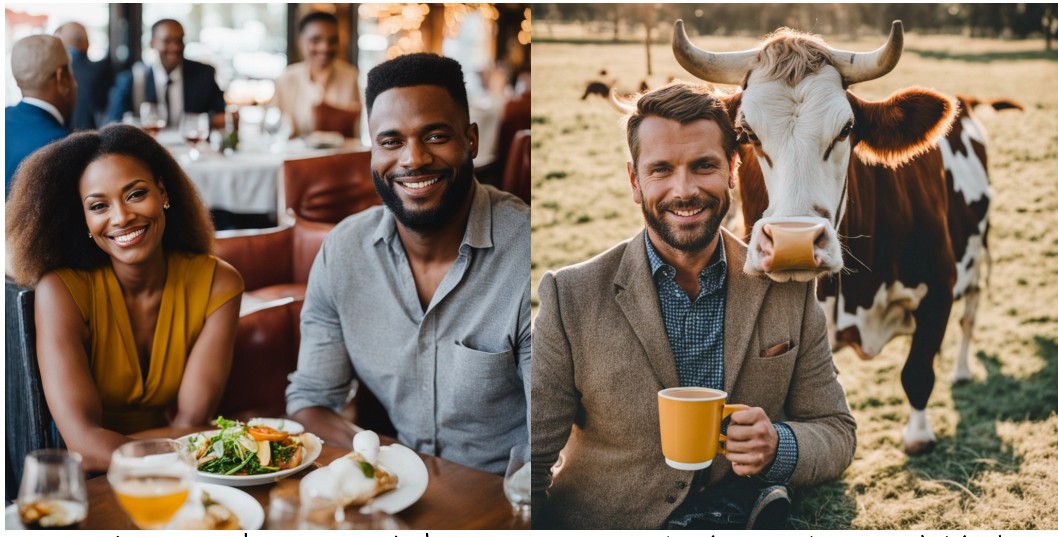

A man and woman seated
at a table in a restaurant.

A cow laying on the grass behind
a man holding a cup of coffee.

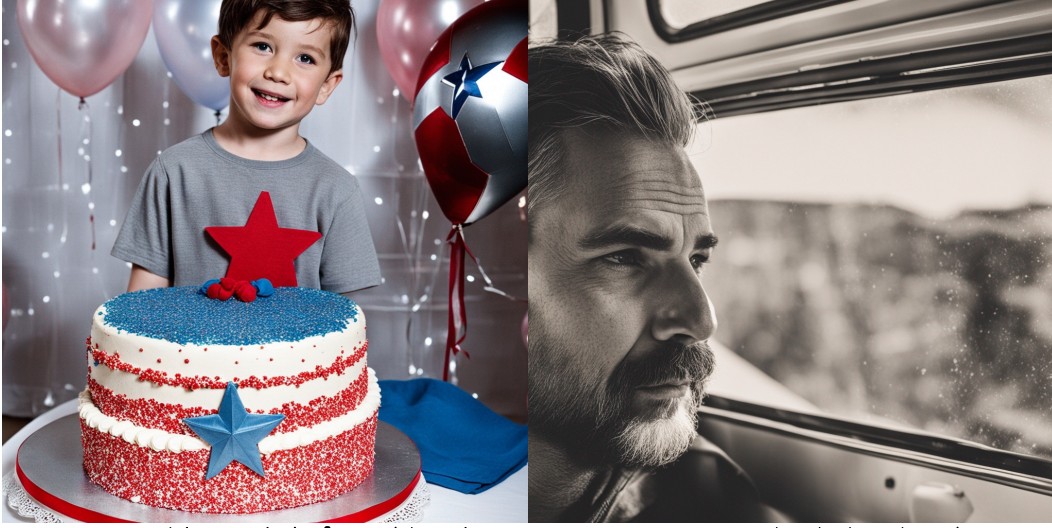

A young kid stands before a birthday
cake decorated with captain America.

A man who is sitting in a bus
looking away from the window.

Figure 17: **Additional Qualitative Results on Zero-Shot MS-COCO Validation.**

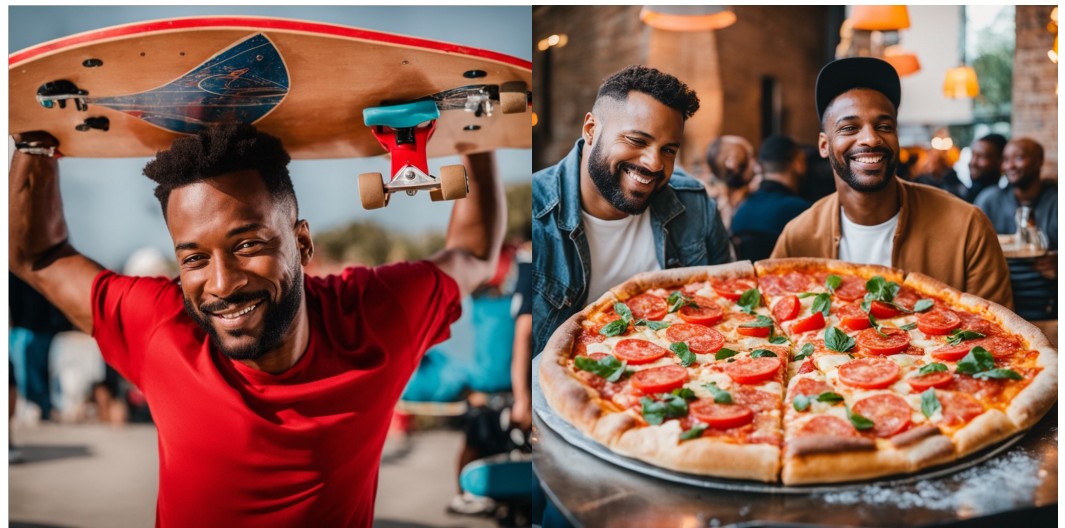

A man in a red shirt is holding a skate board up over his head.

Two men who are sitting next to each other with a large pizza in front of them.

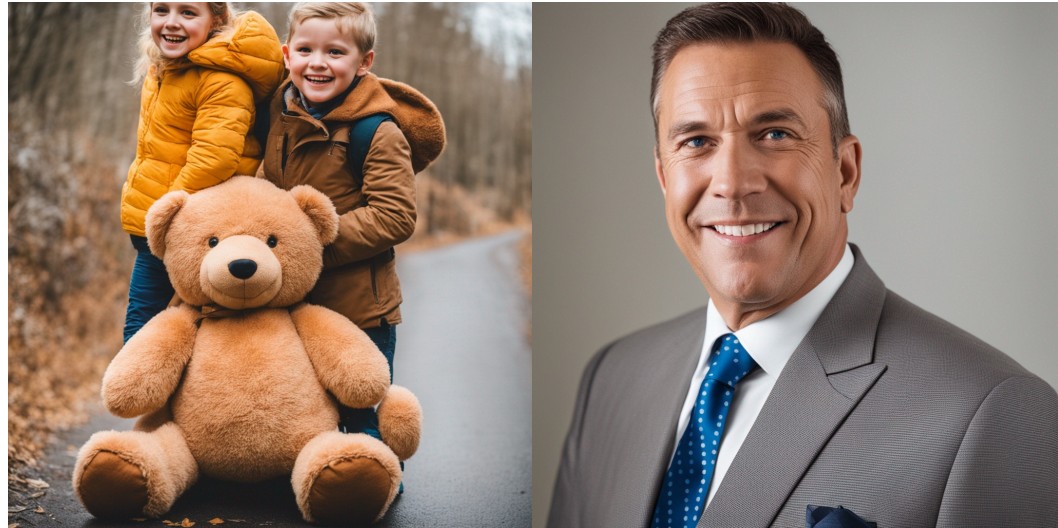

Two children carry an enormous stuffed teddy bear.

The upper half of a man posing for a photograph wearing a suit with a blue tie and matching pocket corner.

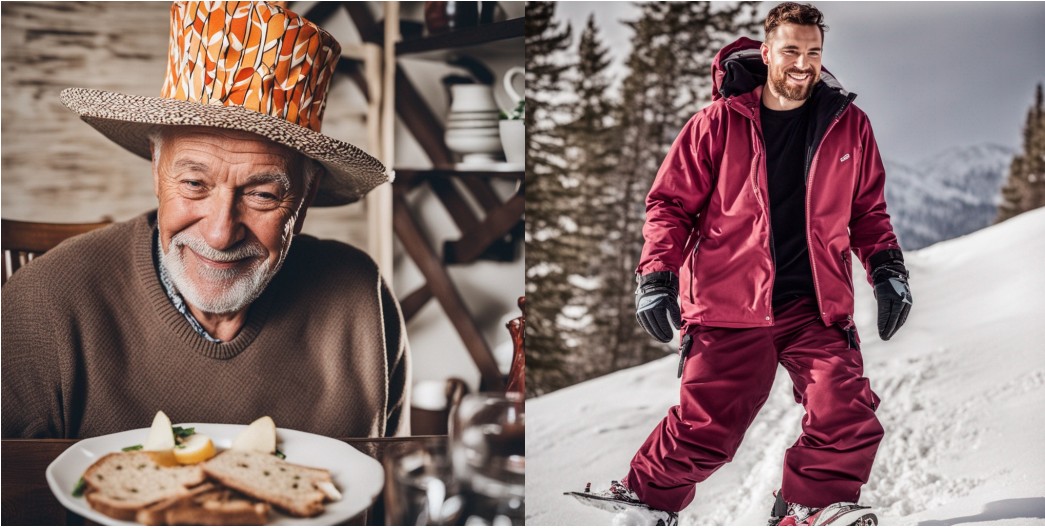

An older man is wearing a funny hat in his dining room.

Young man on top of a snowboard wearing maroon jacket.

Figure 18: **Additional Qualitative Results on Zero-Shot MS-COCO Validation.**

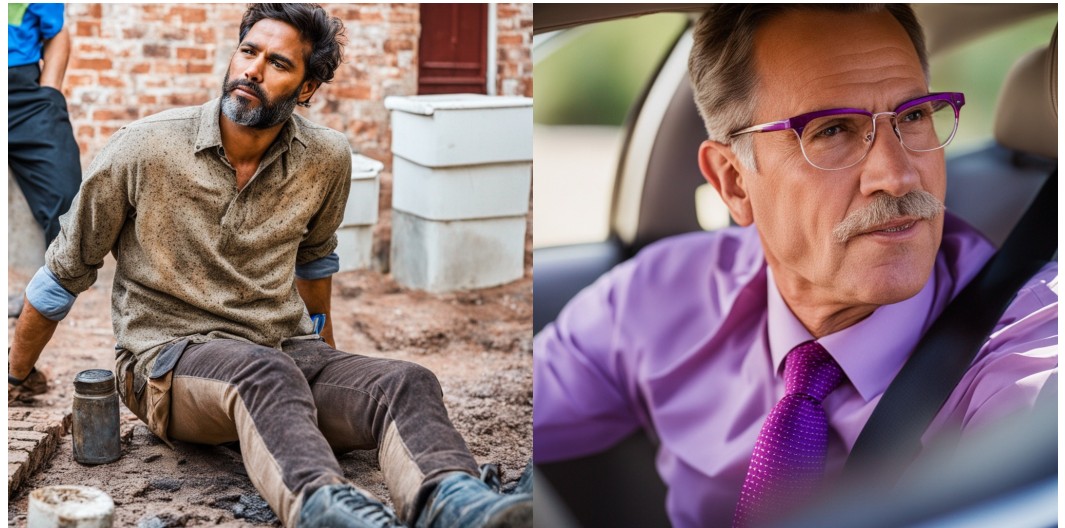

Man sitting on brick covered ground, appearing dirty and tired.

A man wearing a purple neck tie and glasses while sitting in a car.

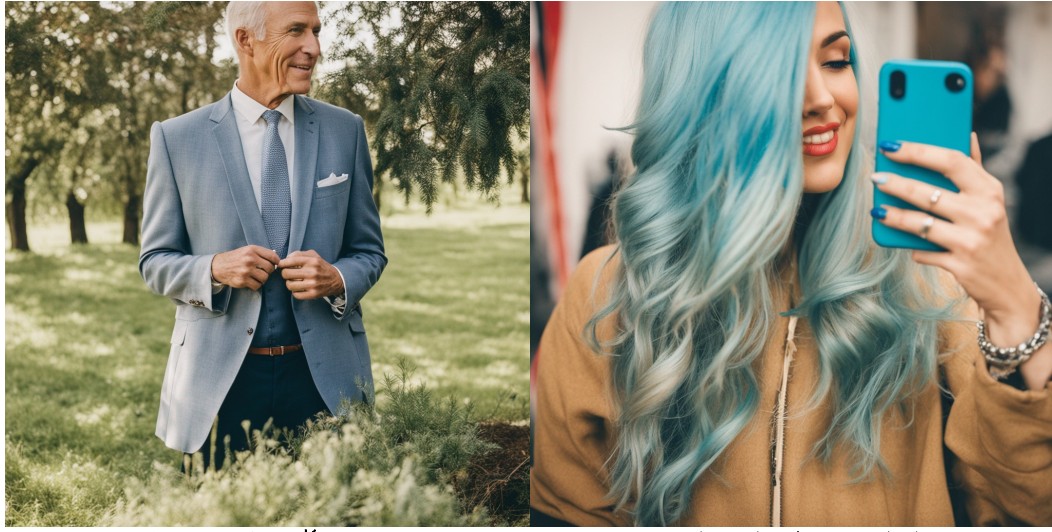

A man standing on grassy area next to trees.

A girl with blue hair is taking a self portrait.

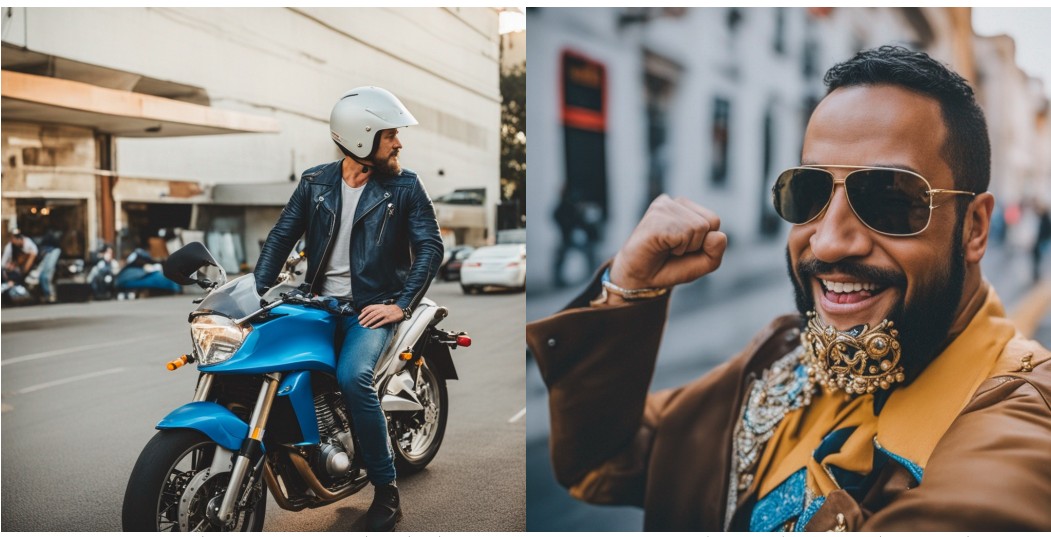

A man wearing a helmet is sitting on his blue motorcycle.

A person dressed up taking a picture at a street with his fist up.

Figure 19: **Additional Qualitative Results on Zero-Shot MS-COCO Validation.**

## A.17   LICENSES

Image Datasets:

- LAION-5B[3] (Schuhmann et al., 2022): Creative Common CC-BY 4.0 license.
- COYO-700M[4] (Byeon et al., 2022): Creative Common CC-BY 4.0 license.
- MS-COCO[5] (Lin et al., 2014): Creative Commons Attribution 4.0 License.

Pretrained Models and Off-the-Shelf Annotation Tools:

- diffusers[6] (von Platen et al., 2022): Apache 2.0 License.
- CLIP[7] (Radford et al., 2021): MIT License.
- Stable Diffusion[8] (Rombach et al., 2022): CreativeML Open RAIL++-M License.
- YOLOS-Tiny[9] (Fang et al., 2021): Apache 2.0 License.
- BLIP2[10] (Guo et al., 2023): MIT License.
- MMPose[11] (Contributors, 2020): Apache 2.0 License.
- ViTPose[12] (Xu et al., 2022): Apache 2.0 License.
- Omnidata[13] (Eftekhar et al., 2021): OMNIDATA STARTER DATASET License.
- MiDaS[14] (Ranftl et al., 2022): MIT License.
- clean-fid[15] (Parmar et al., 2022): MIT License.
- SDv2-inpainting[16] (Rombach et al., 2022): CreativeML Open RAIL++-M License.
- SDXL-base-v1.0[17] (Podell et al., 2023): CreativeML Open RAIL++-M License.
- Improved Aesthetic Predictor[18]: Apache 2.0 License.

---

[3] https://laion.ai/blog/laion-5b/
[4] https://github.com/kakaobrain/coyo-dataset
[5] https://cocodataset.org/#home
[6] https://github.com/huggingface/diffusers
[7] https://github.com/openai/CLIP
[8] https://huggingface.co/stabilityai/stable-diffusion-2-base
[9] https://huggingface.co/hustvl/yolos-tiny
[10] https://huggingface.co/Salesforce/blip2-opt-2.7b
[11] https://github.com/open-mmlab/mmpose
[12] https://github.com/ViTAE-Transformer/ViTPose
[13] https://github.com/EPFL-VILAB/omnidata
[14] https://github.com/isl-org/MiDaS
[15] https://github.com/GaParmar/clean-fid
[16] https://huggingface.co/stabilityai/stable-diffusion-2-inpainting
[17] https://huggingface.co/stabilityai/stable-diffusion-xl-base-1.0
[18] https://github.com/christophschuhmann/improved-aesthetic-predictor

