# OpenReview forum: "HyperHuman: Hyper-Realistic Human Generation with Latent Structural Diffusion"
_ICLR.cc/2024/Conference — ICLR 2024 poster_

### Official Review · Reviewer_jWKR · 2023-10-28

**Soundness:** 3 good
**Presentation:** 3 good
**Contribution:** 3 good
**Rating:** 6
**Confidence:** 3

**Summary:**

This work proposes a framework aiming to better preserve and enhance human generation using T2I models. It has multiple contributions, it first collects a large-scale human-century dataset with structure information like depth/skeleton and surface-normal. It also proposes a latent structure module to jointly model images and other structure output, further, it proposes a structurally guided Refiner to better compose the predicted conditions and improve image quality. Experiments show HyperHuman can achieve SoTA performance in diverse scenarios.

**Strengths:**

– This paper is well written and has clear motivation – trying to improve the control consistency in human generation, which is missing from the current control pipeline work like controlnet and T2I-adapter
– The proposed expert branches are interesting and well-study, i.e. how to balance feature sharing and model capacity for each modality,

**Weaknesses:**

– A study on modality-specific reconstruction using RGB VAE is missing, I think the author needs to study if RGB VAE is suitable to encode structure information or if further finetuning is needed.
-- An important baseline that finetunes on HumanVerse images without structure outputs using the SoTA network architecture is missing. It is not clear to me how much improvement the structure prediction brings to the performance. i.e. do we really need to predict the structure information or we can just fine-tune the general-purpose T2I model to the human images dataset?
-- The curated HumanVerse dataset is claimed to be one of the contributions, but the author did not mention if they will open-source the dataset. I hope the authors can clarify this as it can hurt the contribution of this paper if the dataset is kept internal.

**Questions:**

– It is not clear to me why to use independent noise for modeling the different modalities, have you tried modeling using a single noise?
– Depth, normal etc have different distributions to RGB images, how do you ensure the VAE can capture its semantic in the latent space?
– It is not clear to me how you use the output of the first stage model which is in lower resolution, and input into the Refiner model

---

> ### Author Response · Authors · 2023-11-20
> **Response to Reviewer jWKR [Part 1/2]**
>
> We sincerely thank the reviewer for your insightful comments and recognitions to this work, especially for acknowledging that:
> 1) We curate a large-scale dataset with extensive structural annotations like depth, skeleton, and surface-normal;
> 2) Our latent structural module can jointly model images and other structure output;
> 3) The Structural-Guided Refiner can better compose the predicted conditions and improve image quality;
> 4) Experiments verify that HyperHuman achieves SoTA performance in diverse scenarios;
> 5) The paper is well-written, the motivation is clear to improve control consistency that are missing in existing studies;
> 6) The proposed expert branches are interesting and well-study.
>
> We have polished the paper, added the experiment results and make the clarifications in the revised version. Note that the following polishments have been made according to your advice:
>
> * The modality-specific reconstruction results and analysis are included in the Appendix Section A.14 "VAE Reconstruction Performance on Modality-Specific Input".
>
> Thanks again for your very constructive comments, which have helped us improve the paper quality significantly! Below we would like to provide point-to-point responses to all the raised questions:
>
> > **Q1: "About the study on modality-specific reconstruction using RGB VAE."**
>
> **A1:** Thank you so much for pointing out this problem! We have added below results and clarifications in the Appendix Section A.14 "VAE Reconstruction Performance on Modality-Specific Input":
>
> **1)** We use an improved auto-encoder of the pretrained Stable Diffusion "sd-vae-ft-mse" (https://huggingface.co/stabilityai/sd-vae-ft-mse) as VAE to encode inputs from all the modalities, including RGB, depth, surface-normal, and body skeleton maps. To further validate that RGB VAE can be directly used for other structural maps, we extensively evaluate the reconstruction metrics of all the involved structural maps on $100k$ samples. The results are reported below, which show that the pretrained RGB VAE is robust enough to handle different modality images, including the structural maps we use in this work.
>
> |Modality|$\text{rFID} \downarrow$|$\text{PSNR} \uparrow$|$\text{SSIM} \uparrow$|$\text{PSIM} \downarrow$|
> |-----|:-----:|:-----:|:-----:|:-----:|
> |Body Skeleton|0.49|39.24|0.96|0.188|
> |MiDaS Depth|0.19|47.08|0.99|0.004|
> |Surface-Normal|0.24|40.11|0.97|0.010|
>
> Besides, we additionally show some visualized reconstruction samples in the Figure 12 of the Appendix Section A.14 "VAE Reconstruction Performance on Modality-Specific Input", where in each group, the first row is the input structural maps, and the second row is the reconstructed structural maps from the pretrained RGB VAE. Therefore, both the quantitative metrics and visual results show that the pretrained RGB VAE is robust enough to faithfully reconstruct structural maps.
>
> **2)** It’s a common practive in recent diffusion model studies to encode the structural condition maps with the pretrained Stable Diffusion RGB VAE. For example, in HumanSD [a], they directly use RGB VAE to encode skeleton maps and concatenate with noisy latent for conditioning. In Sketch-Guided Diffusion [b], the authors directly use RGB VAE to encode edge maps.
>
> **3)** At the early stage of our project, we find the pretrained Stable Diffusion RGB VAE robust enough to reconstruct a variety of structural maps, such as MiDaS depth, Omnidata depth, Omnidata surface-normal, canny map, body/face/hand keypoint maps, etc. Besides, we have also compared different structural map encoding methods in a small-scale subset before the final large-scale training, including the pretrained SD VAE, interpolatation (resize) for downsampling, and learnable convolutions in a ControlNet manner. We empirically find the pretrained SD VAE works best in our setting.
>
> > **Q2: The performance of a model trained without structure prediction.**
>
> **A2:** Yes, we totally agree that such a baseline experiment is very important to show the effectiveness of our method. We would like to modestly point out that we have already done this ablation experiment in the "Denoise RGB" setting of Table 2, main paper. This ablation setting means that we only denoise the RGB as target while not simultaneously denoising any structural maps, which just means finetuning on HumanVerse without structure outputs. The performance gain ($20.76$\% in $\text{FID}$ and $23.66$\% in $\text{FID}_\text{CLIP}$) from "Denoise RGB" to the full model ("HyperHuman (Ours)") shows the effectiveness of our proposed method. Besides, such framework also supports more functionalities by jointly outputting the spatially aligned RGB, depth, and surface-normal maps.

---

> ### Author Response · Authors · 2023-11-20
> **Response to Reviewer jWKR [Part 2/2]**
>
> > **Q3: About releasing the dataset.**
>
> **A3:** Thanks again for acknowledging that our dataset could benefit future research! We definitely would love to release the dataset, but are not able to commit to exact timelines at this point.
>
> > **Q4: "It is not clear to me why to use independent noise for modeling the different modalities, have you tried modeling using a single noise?"**
>
> **A4:** Yes, at the early stage of our experiment, we have tried to use a single noise for different modalities. However, the model fails to generate reasonable structural maps and RGB images at the inference stage. The key reason is that sampling a single noise gives the model a shortcut to trivially learn all the modalities. Specifically, when we sample a single noise for different modalities, the required outputs for different branches are exactly the same. In this way, the model can cheat to only learn the distribution of a single modality and just replicating the predicted noise to other branches. Though seemingly converged at the training stage, it fails to capture the joint distribution of all the modalities, which performs poorly in simultaneously generating aligned results.
>
> > **Q5: "Depth, normal etc have different distributions to RGB images, how do you ensure the VAE can capture its semantic in the latent space?"**
>
> **A5: 1)** As also shown in **A1**, even for the depth and surface-normal maps of different distributions to RGB images, they can still be well reconstructed both quantitatively and visually. This suggests that we faithfully compress structural maps like depth and surface-normal into latent space for diffusion learning.
>
> **2)** A faithful reconstruction means there is little compression loss in the structural map encoding process. Such accurate reconstruction emcompasses both the low-level image details and rich latent space semantics.
>
> > **Q6: "It is not clear to me how you use the output of the first stage model which is in lower resolution, and input into the Refiner model."**
>
> **A6:** In the current pipeline, we do not take the low-resolution image $\mathbf{x}$ as a condition for the second-stage Structure-Guided Refiner, mainly due to two considerations:
>
> **1)** It’s hard to take a coarse-version RGB image as condition at the training stage, since it is too time consuming to use the diffusion-based inference to get the coarse images from the first-stage model. An efficient choice is to get a low-resolution image simply by downsampling (directly resize to low-resolution images). However, this will make the second-stage model degrade to a super-resolution model. The disadvantage is that the generation diversity is largely limited, since we can only create a super-resolution view of the original image, while the appearances can not be flexibly changed.
>
> **2)** Taking the predicted structural maps rather than the low-resolution images as conditions strike a good balance between the structural guidance and generation freedom: **a)** On the one hand, they give the fine-grained explicit structural guidance for generation of better quality. **b)** On the other hand, unlike the coarse image condition that restricts the appearance in synthesis, it still leaves much freedom for diverse results. With such conditioning design, our second-stage Structure-Guided Refiner can be inserted into any pretrained diffusion models in a plug-and-play manner.
>
> ****
>
> **References**
>
> [a] - Ju et al. "HumanSD: A Native Skeleton-Guided Diffusion Model for Human Image Generation." ICCV, 2023.
>
> [b] - Voynov et al. "Sketch-Guided Text-to-Image Diffusion Models." SIGGRAPH, 2023.
>
> [c] - Rombach et al. "High-Resolution Image Synthesis with Latent Diffusion Models." CVPR, 2022.
>
> ****
>
> Please don’t hesitate to let us know if there are any additional clarifications or experiments that we can offer!

---

> > ### Comment · Reviewer_jWKR · 2023-11-22
> >
> > After reading the authors' responses, I decided to raise my original rating. I do encourage the author to release HyperHuman dataset as part of the contribution of this paper.

---

> > > ### Author Response · Authors · 2023-11-22
> > > **Thanks for your comments!**
> > >
> > > Dear Reviewer jWKR:
> > >
> > > We are delighted to hear that your concerns are addressed! Many thanks again for your very constructive comments, which have helped us improve the quality of the paper significantly.
> > >
> > > Best,
> > >
> > > Paper 350 Authors.

---

### Official Review · Reviewer_1Ma5 · 2023-10-30

**Soundness:** 3 good
**Presentation:** 3 good
**Contribution:** 2 fair
**Rating:** 8
**Confidence:** 4

**Summary:**

The paper proposes a unified framework called HyperHuman for generating high-quality human images with diverse poses, appearances, and layouts. The authors curate a new dataset called HumanVerse with rich annotations (human pose, depth, and surface normal) to train the model. The framework consists of two modules: the Latent Structural Diffusion Model and the Structure-Guided Refiner. The former module denoises the depth and surface normal along with the RGB image conditioned on the caption and skeleton, while the latter module generates higher-resolution images based on the predicted depth, surface normal, and provided skeleton. The paper's contributions include introducing a new dataset with rich annotations, proposing a novel framework for human image generation which yields superior performance, and generating realistic humans under diverse scenarios.

**Strengths:**

1. The paper proposes a large-scale curated human-centric dataset with comprehensive annotations (human pose, depth map, surface normal) that may benefit a lot in future research in the field of human image generation.
2. The author introduces a new approach for incorporating the body skeletons, depth and surface normals by jointly denoising depth, surface normal and RGB images and a robust conditioning scheme with text prompt and human pose. The paper also conducts ablation study to prove the effectiveness of this.
3. The proposed method is extensively evaluated on the human-centric dataset MS-COCO 2014 Validation Human to many other SOTA approaches and outperforms on almost quantitative metrics.
4. The paper also extensively experiments and does an ablation study on the effectiveness of the Structure-Guided Refiner with and without many conditions, noise scheduler and expert branch denoising mechanism.

**Weaknesses:**

1. The label of the HumanVerse dataset has been created based on other pre-trained models, such as depth and pose estimation models. However, the quality of the dataset does not guarantee accurate labelling (especially in crowded scene or extreme lighting condition), and its reusability for other tasks remains uncertain.
2. The approach has a noteworthy computational cost. However, when evaluating the results in comparison to the second approach, it becomes evident that the gains in Pose Accuracy are not substantial. (Tab. 1)
3. The paper's objective is to address incoherent parts and unnatural poses. However, it falls short in terms of providing quantitative metrics to evaluate the effectiveness of the proposed method in addressing these issues.
4. This paper lacks a fair comparison on two fronts. Firstly, it compares itself to other non-pose conditional guided methods (such as SD and DeepFloy-IF). Secondly, the method discussed in the paper was trained on the HumanVerse dataset, which exclusively contains the rich human class, while other methods are trained on a broader set of classes.

**Questions:**

1. According to Figure 2, the pose skeleton is encoded before being concatenated with noise. What is the encoder used for the pose, and how does the author process the pose input before encoding it?
2. What about inference requirements (GPU memory, time)?
3. How do we ensure the input pose and caption are aligned with the predicted depth, map, and surface normal? Is there any evaluation on this?
4. In the qualitative result, the comparison among other SOTA methods does not include the pose skeleton figure, I suggest adding it for better visualization.
5. The description of Figure 2 lacks clarity, particularly when the authors refer to the colours "purple" and "blue." However, the figure itself makes it confusing to recognize which part they mention. I suggest the authors should redraw it.
6. I suggest adding the ablation study between joint denoising the targets and individually denoising the targets to show the effectiveness of joint denoising. This is the main technical contribution beside the new dataset.
7. In Table 7: Additional Ablation Results for Structure-Guided Refiner, no ablation study on conditioning on predicted low-resolution image x (from the first module).

**Details Of Ethics Concerns:**

This paper collects many human faces and images. Thus, privacy concern is raised.

---

> ### Author Response · Authors · 2023-11-20
> **Response to Reviewer 1Ma5 [Part 1/4]**
>
> We sincerely thank the reviewer for your insightful comments and recognitions to this work, especially for acknowledging that:
> 1) We introduce a large-scale dataset with rich annotations to benefit future human generation research;
> 2) We propose a novel framework with superior performance;
> 3) We can generate realistic humans under diverse scenarios;
> 4) The effectiveness of jointly denoising RGB, depth, and normal is proven by ablation study;
> 5) We extensively evaluate on human-centric COCO dataset and outperform many SOTA baselines;
> 6) Each approach design, including Structure-Guided Refiner w/ and w/o many conditions, noise scheduler and expert branch denoising mechanism, are extensively verified by experiments and ablation studies.
>
> We have polished the paper, added the experiment results and make the clarifications in the revised version. Note that the following polishments have been made according to your advice:
>
> * The clarifications on pose processing and encoding details are made in the Appendix Section A.13 "More Details on Pose Processing and Encoding".
>
> * The inference requirements are added in the "Implementation Details" paragraph of the main paper Section 5.
>
> * The pose skeleton figures for all the qualitative comparisons are added in the Figure 13, 14, 15, 16 of the Appendix Section A.13 "More Details on Pose Processing and Encoding".
>
> * The pipeline illustration is polished in Figure 2 of the main paper to make it easier recognizable.
>
> Thanks again for your very constructive comments, which have helped us improve the paper quality significantly! Below we would like to provide point-to-point responses to all the raised questions:
>
> > **Q1: "The label of the HumanVerse dataset has been created based on other pre-trained models, such as depth and pose estimation models. However, the quality of the dataset does not guarantee accurate labelling (especially in crowded scene or extreme lighting condition), and its reusability for other tasks remains uncertain."**
>
> **A1: 1)** It’s a common practice for existing studies to use other pre-trained models for estimation and labeling. For example, SDv2-depth (https://huggingface.co/stabilityai/stable-diffusion-2-depth) also uses MiDaS [a] for depth labeling; MonoSDF [b] also uses Omnidata [c] for surface-normal estimation as supervision and structural guidance; ControlNet, T2I-Adapter, and HumanSD also use OpenPose [d] or ViTPose [e] for body keypoint detection.
>
> **2)** We have tried our best to make the annotations more accurate. For example, when choosing the backbone for body skeleton annotation, we use ViTPose-H, which performs the best over several pose estimation benchmarks. Besides, we also propose to outpaint each image for a more holistic, which gives more accurate annotations. Please kindly refer to the "Outpaint for Accurate Annotations" paragraph of main paper Section 4 for how we improve annotation pipelines for more accurate labeling.
>
> **3)** We admit that certain estimation error indeed exists due to the limitation of current state-of-the-art estimators. However, training on the large-scale dataset with noisy labelings can still contribute to a robust model. For example, the image captions of current text-to-image dataset (*e.g.*, LAION and COYO) are quite noisy, with some irrelevant information like HTTP tags or random emojis. In spite of this, robust text-to-image models can be well trained with unprecedented performance and satisfactory text-image alignment.
>
> > **Q2: "About the Pose Accuracy gain in Table 1."**
>
> **A2: 1)** We would like to modestly point out that all the controllable baselines of ControlNet, T2I-Adapter, and HumanSD sacrifice image quality and diversity for better pose accuracy. As can be seen in both quantitative evaluation metrics and qualitative visual comparisons, our model can generate more realistic humans of better quality and richer diversity.
>
> **2)** As reported in Table 5 of the "Pose Accuracy Results on Different CFG Scales" paragraph in Appendix Section A.1, we are consistently better than controllable T2I baselines in terms of pose accuracy, over multiple CFG scale ranging from $4.0$ to $13.0$. This suggests that we can make the best of two worlds by giving both pose-aligned and high-quality human generation results.

---

> ### Author Response · Authors · 2023-11-20
> **Response to Reviewer 1Ma5 [Part 2/4]**
>
> > **Q3: "About evaluating the incoherent parts and unnatural poses."**
>
> **A3: 1)** The incoherent parts and unnatural poses appear as human image artifacts in the pixel space. Since the image quality metrics of $\text{FID}$, $\text{KID}$, and $\text{FID}_\text{CLIP}$ calculate the distribution distance between the real data and generated data, they can capture those generated artifacts that have gap with real data distribution. In this way, the incoherent parts and unnatural poses can be reflected with poor image quality metrics. The superior performance of our method on image quality metrics can show the effectiveness for solving these unnatural cases.
>
> **2)** Though incoherent parts and unnatural poses could be hard to quantitatively evaluated with automatic pipeline, we have conducted a comprehensive user study with extensive visual comparisons. As human evaluators are highly sensitive to the human image artifacts of unreasonable structures, the better human preference of our model shows our effectiveness. Besides, from the extensive qualitative comparisons as shown in the main paper Figure 1 and Appendix Figure 13, 14, 15, 16, we surpass recent state-of-the-art T2I models with more coherent and natural human generation.
>
> > **Q4: "Comparison with some non-pose conditional guided methods, and the fact that not all existing works are trained on the exact same dataset."**
>
> **A4: 1)** We would like to modestly re-emphasize that our focused setting is **pose-conditioned human generation**. Human generation is one of the most challenging sub-domains in current text-to-image studies, which fails to be well addressed in general T2I models. Therefore, we consider it as an individual research problem.
>
> * Based on this, we follow the comparison settings of previous controllable text-to-human studies to compare with both general T2I models and controllable methods with pose condition, and use the officially released model for all the baselines. We would like to politely point out that the amount of resources and computation to re-train all existing large-scale foundation image models on the same dataset would be too huge for us to afford. We sincerely hope that you could kindly understand this.
>
> * For comprehensive comparisons, we have grouped comparison models into two categories (the "Comparison Methods" paragraph of main paper Section 5): **a)** General T2I models, including SDv1.5, SDv2.0, SDv2.1, SDXL, and DeepFloyd-IF; **b)** Controllable methods with pose condition, including ControlNet, T2I-Adapter, and HumanSD. To further highlight the difference between these two types of comparison methods, we have modified the table by adding a horizontal line to make them more clearly separated. Please kindly refer to Table 1 of the main paper for the revision. We would like to kindly mention that the reason we include the results from non-posed conditional models like the Stable Diffusion series is to provide *a detailed and comprehensive analysis* between various approaches.
>
> **2)** All the quantitative and qualitative evaluations are conducted on the zero-shot MS-COCO 2014 validation dataset, which is unseen to all the tested models. This can further guarantee the fairness of comparisons.
>
> > **Q5: "According to Figure 2, the pose skeleton is encoded before being concatenated with noise. What is the encoder used for the pose, and how does the author process the pose input before encoding it?"**
>
> **A5:** Thank you for pointing out this question! We have polished to make clarifications in the Appendix Section A.13 "More Details on Pose Processing and Encoding".
>
> **1)** The encoder used for pose is the pretrained VAE encoder of Stable Diffusion, which is the same as the encoder used for RGB, depth, and surface-normal maps.
>
> **2)** Before pose encoding, we visualize the body keypoints on a black canvas to form a skeleton map, similar to previous controllable methods with pose condition. Specifically, we use exactly the same pose drawing method as HumanSD and T2I-Adapter to ensure fairness.
>
> > **Q6: "What about inference requirements (GPU memory, time)?"**
>
> **A6:** We run inference on a single 40GB NVIDIA A100 GPU. The first-stage Latent Structural Diffusion Model uses 2 seconds for 50-step DDIM sampling, which is roughly similar to SDv2.0. The second-stage Structure-Guided Refiner uses 10 seconds for 50-step DDIM sampling, which is roughly similar to SDXL. We have included this in the "Implementation Details" paragraph of the main paper Section 5.

---

> ### Author Response · Authors · 2023-11-20
> **Response to Reviewer 1Ma5 [Part 3/4]**
>
> > **Q7: "How do we ensure the input pose and caption are aligned with the predicted depth, map, and surface normal? Is there any evaluation on this?"**
>
> **A7: 1)** We have evaluated the spatial alignment between the predicted RGB and structural maps (depth and surface-normal maps) in ablation study. Specifically, we extract the depth and surface-normal from the predicted RGB image by off-the-shelf estimator, then calculate the $\mathcal{L}_2$-error with the predicted depth and surface-normal as a proxy for alignment. The ablation study results are shown in Table 2 of the main paper, which shows that our structural expert branch design can generate aligned RGB images, depth and surface-normal maps. Besides, the pose-images and RGB images are aligned as validated by Pose Accuracy metrics, and the image and captions are aligned as validated by text-image alignment metrics of CLIP score. Therefore, we can conclude that the input pose, text prompt (caption), synthesized RGB image, predicted depth and surface-normal maps are all aligned with each other.
>
> **2)** Since we can use the predicted depth and surface-normal to generate the aligned image with input pose and caption, this indirectly shows that they are aligned. Otherwise, if there exists misalignment, the generated image will be unaligned to input pose or caption, leading to generation conflicts. Extensive visualization samples also show that the predicted depth and surface-normal maps are aligned with input pose and caption.
>
> > **Q8: "In the qualitative result, the comparison among other SOTA methods does not include the pose skeleton figure, I suggest adding it for better visualization."**
>
> **A8:** Thanks for your precious advice! The Figure 1 of the main paper shows the corresponding conditioning pose for each generation sample. We have further included the pose skeleton figures for all the qualitative comparisons in the Figure 13, 14, 15, 16 of the Appendix Section A.13 "More Details on Pose Processing and Encoding".
>
> > **Q9: "Improve the description of Figure 2."**
>
> **A9:** Many thanks for your advice! We have polished the pipeline illustration in Figure 2 of the main paper to make it easier recognizable.
>
> > **Q10: "I suggest adding the ablation study between joint denoising the targets and individually denoising the targets to show the effectiveness of joint denoising."**
>
> **A10:** Thank you so much for the insightful suggestion!
>
> **1)** Actually, we have tried to individually denoise each target at the early stage of this work, where we separately finetune the text2depth, text2normal, and text2image models. However, we find that due to the high-variance and stochaticity of diffusion model, the individually predicted RGB, depth, and surface-normal are totally unaligned with each other. Even with the help of same prompt, same random seed, and same initial noise $\mathbf{x}_T$, they are still mismatched. This makes them unable to be used for the second-stage Structure-Guided Refiner of conditional generation.
>
> **2)** Due to the current resource and time limit, we could not conduct this ablation experiment right now. But we will definitely add the experiment results in the final version. Thanks in advance for your kind understanding to this situation!
>
> > **Q11: "In Table 7: Additional Ablation Results for Structure-Guided Refiner, no ablation study on conditioning on predicted low-resolution image x (from the first module)."**
>
> **A11:** In the current pipeline, we do not take the low-resolution image $\mathbf{x}$ as a condition for the second-stage Structure-Guided Refiner, mainly due to two considerations:
>
> **1)** It’s hard to take a coarse-version RGB image as condition at the training stage, since it is too time consuming to use the diffusion-based inference to get the coarse images from the first-stage model. An efficient choice is to get a low-resolution image simply by downsampling (directly resize to low-resolution images). However, this will make the second-stage model degrade to a super-resolution model. The disadvantage is that the generation diversity is largely limited, since we can only create a super-resolution view of the original image, while the appearances can not be flexibly changed.
>
> **2)** Taking the predicted structural maps rather than the low-resolution images as conditions strike a good balance between the structural guidance and generation freedom: **a)** On the one hand, they give the fine-grained explicit structural guidance for generation of better quality. **b)** On the other hand, unlike the coarse image condition that restricts the appearance in synthesis, it still leaves much freedom for diverse results. With such conditioning design, our second-stage Structure-Guided Refiner can be inserted into any pretrained diffusion models in a plug-and-play manner.

---

> ### Author Response · Authors · 2023-11-20
> **Response to Reviewer 1Ma5 [Part 4/4]**
>
> > **Q12: "Ethical issues of Responsible research practice (e.g., human subjects, data release). Privacy concern of human face and images."**
>
> **A12:** Thank you for pointing out the potential ethical issues!
>
> **1)** We are also aware of the potential negative impacts under the misuse of this technique. **The intention of this work is to benefit the research community and applications**. In view of this, we have provided our ethical considerations in Appendix Section A.6.
>
> **2)** We believe that the proper use of this technique will enhance the machine learning research and digital entertainment. We provide some additional measures to avoid negative social impacts:
>
> * We can add a visible or invisible watermark to the generated images to make it clear that they are not real photographs of real people. This can help prevent malicious users from using the images to deceive or manipulate others;
>
> * We can consider limiting access to the generated images to only authorized users, such as researchers or professionals in the field. This can help prevent the general public from misusing the images;
>
> * We can clearly state the terms of use for the generated images, including restrictions on how they can be used and distributed. This can help prevent malicious users from using the images for harmful purposes;
>
> * We can develop ethical guidelines for the use of the generated images, and require all users to adhere to these guidelines. This can help ensure that the images are used in a responsible and ethical manner.
>
> ****
>
> **References**
>
> [a] - Ranftl et al. "Towards Robust Monocular Depth Estimation: Mixing Datasets for Zero-shot Cross-dataset Transfer." T-PAMI 2022.
>
> [b] - Yu et al. "MonoSDF: Exploring Monocular Geometric Cues for Neural Implicit Surface Reconstruction." NeurIPS 2022.
>
> [c] - Eftekhar et al. "Omnidata: A Scalable Pipeline for Making Multi-Task Mid-Level Vision Datasets from 3D Scans." ICCV 2021.
>
> [d] - Cao et al. "OpenPose: Realtime Multi-Person 2D Pose Estimation using Part Affinity Fields." CVPR 2017.
>
> [e] - Xu et al. "ViTPose: Simple Vision Transformer Baselines for Human Pose Estimation." NeurIPS 2022.
>
> ****
>
> Please don’t hesitate to let us know if there are any additional clarifications or experiments that we can offer!

---

> ### Comment · Reviewer_1Ma5 · 2023-11-22
>
> I would like to thank the authors for their response. The response has addressed most of my concerns. Regarding the quality of the annotations of the datasets, I would suggest training pose, depth....models on the proposed annotation and evaluating it on the standard test sets of some common datasets to see the performance compared with the same models trained on the ground-truth annotation. This will justify the quality of the generated annotations. I would like to increase my score to 8.

---

> > ### Author Response · Authors · 2023-11-23
> > **Thanks for your comments!**
> >
> > Dear Reviewer 1Ma5:
> >
> > We are delighted to hear that your concerns are addressed! Thanks for your precious advice about further evaluations on the annotation quality, we will add these parts in the final version. Many thanks again for your very constructive comments, which have helped us improve the quality of the paper significantly.
> >
> > Best,
> >
> > Paper 350 Authors.

---

### Official Review · Reviewer_HmdV · 2023-11-01

**Soundness:** 3 good
**Presentation:** 3 good
**Contribution:** 3 good
**Rating:** 6
**Confidence:** 4

**Summary:**

The paper constructs a large-scale dataset HumanVersea containing 340M images for better human image generation with stable diffusion. The proposed approach also denoised depth and surface normal in addition to the RGB space and shows further improvements. The paper compares with multiple baselines to demonstrate the superiority of the proposed method.

**Strengths:**

1. The proposed approach is simple yet effective. It’s interesting to see how predicting depth and normal improves the human image quality.

2. The paper addresses an important problem where existing methods show various limitations. The paper compares multiple baseline methods and shows promising results. The constructed dataset is a notable contribution.

3. The paper is in general easy to read.

**Weaknesses:**

1. The curated dataset is filtered with the criteria: only those images containing 1 to 3 human bounding boxes are retained; people should be visible with an area ratio exceeding 15%; plus rule out samples of poor aesthetics (< 4.5) or low resolution (< 200 × 200). I wonder if these rules reduce the diversity of the images that the model could produce, e.g., group/family photo or small faces? This is also observed in some visualizations in the supplementary materials where it does seem like the model improves the visual quality at the cost of diversity. I wonder if the authors could comment more on this? Besides, the CLIP alignment score is also a bit lower than SDXL, is it related to this diversity issue?

2. Since providing accurate poses in real applications is hard, it would be very useful if the model also produces pleasant unconditional results. So I am also curious about the unconditional results. To be specific, how about images synthesized with just the text input without the poses?

3. Related to the previous question, how robust the model is to the input pose? If the input skeleton is jittered for all joints, would the model still produce high-quality images? I am also curious if the sampled noise is fixed and only the input pose is animated, is it possible to animate the image?

4. In Table 2, how about RGB + normal? Since depth and normal are closely related, is it necessary to include both?

5. Regarding the results in Table 1, I am not sure why SDXL is much worse than SD? Could the authors further clarify on this? In addition, it also looks a bit strange to have DeepFloyd-IF 1024 images downsampled. Is it possible to directly compare the 1024 resolution with the proposed method?

6. Since the numerical results are mainly without the second-stage refiner plus the second-stage dataset is internal, it would be helpful to show visual results with only the first stage.

7. I am not sure if this intuition is well explained. Maybe the authors could further clarify “Such monotonous images may leak low-frequency signals like the mean of each channel during training; The zero terminal SNR (αT = 0, σT = 1) is further enforced to eliminate structure map’s low-frequency information.” in subsection “Noise Schedule for Joint Learning”.

8. Will the dataset be released for both stages?

9. Does the input pose also contain the face landmarks or hand poses? Or just the body skeleton? If it's just skeleton, would adding hand poses helps with the generated details?

**Questions:**

Pease see my questions above. I think in general the paper shows promising results but I am not sure if this comes with additional cost of the diversity.

---

> ### Author Response · Authors · 2023-11-20
> **Response to Reviewer HmdV [Part 1/4]**
>
> We sincerely thank the reviewer for your insightful comments and recognitions to this work, especially for acknowledging that:
> 1) Our approach to additionally denoise depth and surface-normal shows further improvements;
> 2) We compare with multiple baselines to demonstrate the superiority with promising results;
> 3) The method is simple yet effective with interesting finding that extra structural map prediction improves human image quality;
> 4) We address an important problem where existing methods show various limitations;
> 5) The constructed large-scale dataset is a notable contribution;
> 6) The paper is in general easy to read.
>
> We have polished the paper, added the experiment results and make the clarifications in the revised version. Note that the following polishments have been made according to your advice:
>
> * The unconditional generation results and analysis are included in the Appendix Section A.9 "Model Performance without Input Pose".
>
> * The jittered pose and animation experiment results and analysis are included in the Appendix Section A.10 "Model Performance on Jittered Pose and Image Animation".
>
> * The evaluation results for "Denoise RGB + Normal" ablation setting are added to Table 2 of the main paper.
>
> * More first-stage visual results are included in the Appendix Section A.10 "More First-Stage Generation Results".
>
> * The Clarifications of updated noise schedule are added in the Appendix Section A.12 "Detailed Intuition of Updated Noise Schedule".
>
> Thanks again for your very constructive comments, which have helped us improve the paper quality significantly! Below we would like to provide point-to-point responses to all the raised questions:
>
> > **Q1: "How the data filtering criteria relates to the generation diversity. The comparison with SDXL for CLIP score."**
>
> **A1: 1)** Filtering out low-resolution or poor-aesthetic images for training is a common practice in prevalent diffusion text-to-image pipelines, *e.g.*, SDv1.5 and SDv2.0.
>
> **2)** We filter out too-many persons and over-small human images mainly due to below reasons:
>
> * For those small human images, the prompts are very likely to be uncorrelated with human, since people is not the main subject of this image. This could confuse the model training.
>
> * For the too-many person images, there are two issues with its caption and image quality: **a)** The noisy image captions mostly fail to describe the detailed information of each person, forcing the model to learn an one-to-many mapping from prompts like “a group of people” to *arbitrary number* of *arbitrary appearance* humans, which is an ill-mannered problem and hard to learn. **b)** Most multi-person images are confronted with the camera focus problem, where camera focusing on the nearer or further humans will make other human instances highly blurry of bad quality.
>
> * The focused problem of this work is pose-conditioned human generation, which allows users control body skeleton according to their needs. Therefore, we follow the experiment settings of controllable T2I baselines like ControlNet, T2I-Adapter and HumanSD to filter an image subset that could be accurately annotated by off-the-shelf pose estimators. Notably, our dataset domain is much more **diverse** than baselines, including various backgrounds and partial human regions such as clothing and limbs.
>
> **3)** The image quality evaluation metrics of $\text{FID}$, $\text{KID}$, and $\text{FID}_\text{CLIP}$ calculate the distribution gap between the real data and generated data, which can reflect both aspects of image quality and **diversity** [a]. The superior performance on all the image quality metrics can reflect the diversity of our method. As also recognized by **Reviewer zFqJ, 1Ma5, jWKR**, we can generate high-quality human images in **diverse** scenarios.
>
> **4)** As shown in Figure 13, 15, 16, 17, 18 in the Appendix, we manage to generate multiple humans of high realism and image quality. Besides, throughout all the qualitative results and visual comparisons, we have demonstrated diverse generation of various layouts under diverse scenarios, *e.g.*, different age groups of baby, child, young people, middle-aged people, and old persons; different contexts of canteen, indoor scenes, in-the-wild roads, snowy mountains, and streetview, etc. How to enhance realistic multi-person generation of high diversity remains an open question. Thank you so much for pointing out this important problem! We will explore this in future work.
>
> **5)** We would like to modestly point out that CLIP score focuses more on text-image alignment, rather than image diversity. Besides, as explained in the "Quantitative Analysis" paragraph of main paper Section 5.1, SDXL uses two text encoders with $3\times$ larger UNet of more cross-attention layers, leading to superior text-image alignment. In spite of this, we still obtain an on-par CLIP score and surpass all the other baselines that have similar text encoder parameters.

---

> ### Author Response · Authors · 2023-11-20
> **Response to Reviewer HmdV [Part 2/4]**
>
> > **Q2: "How about images synthesized with just the text input without the poses?"**
>
> **A2:** Thank you so much for pointing out this! We have included the unconditional generation results and analysis in the Appendix Section A.9 "Model Performance without Input Pose". Thanks to our framework design of robust conditioning scheme, the model is trained to predict reasonable denoising results, even when the conditions are dropout or masked. Therefore, we manage to create realistic human images with superior performance even without the pose skeleton as input.
>
> > **Q3: "Generation results with jittered skeleton joints. Animated images with fixed noise but changed pose."**
>
> **A3:** We sincerely thank you for the insightful comments! We have included the jittered pose and animation experiment results and analysis in the Appendix Section A.10 "Model Performance on Jittered Pose and Image Animation" according to your constructive advice:
>
> **1)** We show additional results on the jittered human poses in Figure 9 of the Appendix Section A.10 "Model Performance on Jittered Pose and Image Animation". Specifically, we first condition on the original pose skeleton and obtain the generated image based on text prompt "A woman standing near a lake with a snow capped mountain behind'' (sub-figure **(a)** and **(b)** in Figure 9). Then we gradually add Gaussian noise to all the joints, from the sigma scale of $2.5$ to $12.5$. It can be seen that our model could produce pleasant results under Gaussian noises to all joints, creating highly pose-aligned images.
>
> **2)** This is a very interesting question! To further verify if we can animate a certain image by gradually changing the input pose, we fix the random seed, the initial starting noise $\mathbf{x}_T$, and text prompt. The sequential generation results are shown in Figure 10 of the Appendix Section A.10 "Model Performance on Jittered Pose and Image Animation". Note that we fix the text prompt of ``A woman standing near a lake with a snow capped mountain behind''. The input skeleton are shifted towards the right side, each step by 10 pixels. Even though we maintain other conditions fixed, we can still see background and appearance changes. We regard this as a promising research problem and will explore it in future work.
>
> > **Q4: "In Table 2, how about RGB + normal? Since depth and normal are closely related, is it necessary to include both?"**
>
> **A4: 1)** Thank you for pointing out this missing ablation study and apologize for forgetting to include this result in main paper. We have revised to add the evaluation results for "Denoise RGB + Normal" ablation setting in Table 2 of the main paper, which are as follows:
>
> |Ablation Settings|$\text{FID} \downarrow$|$\text{FID}_\text{CLIP} \downarrow$|$\mathcal{L}_2^{\mathbf{d}} \downarrow$|$\mathcal{L}_2^{\mathbf{n}} \downarrow$|
> |:-----:|:-----:|:-----:|:-----:|:-----:|
> |Denoise RGB + Normal|19.24|9.15|-|130.6|
> |**HyperHuman (Ours)**|**17.18**|**7.82**|**502.1**|**121.6**|
>
> **2)** Yes, we totally agree that depth and normal are closely related to each other! But we would like to modestly point out that they each contain modality-specific information, which can be further used to complement to each other:
>
> * Depth maps focus more on the spatial relationship, where the depth value difference between the foreground human and background could be quite large. However, the detailed shape variance can hardly be depicted, with almost the same value within an object’s local region (as shown in the depth maps in 2x2 grid of the main paper Figure1);
>
> * Surface-normal maps focus more on the normal direction of object surface, which can better depict the object geometry information. However, it fails to tell which object is nearer or further to the camera (as shown in the surface-normal maps in 2x2 grid of the main paper Figure1).
>
> **3)** The effectiveness of simultaneously adding depth and normal as extra learning targets has been verified in ablation study (Table 2 of the main paper), where the numerical results of full model (HyperHuman, Denoise RGB + Depth + Normal) are better than only use depth (Denoise RGB + Depth) and only use normal (Denoise RGB + Normal). The results suggest that simultaneously adding both depth and normal is beneficial, better than only use one of them.

---

> ### Author Response · Authors · 2023-11-20
> **Response to Reviewer HmdV [Part 3/4]**
>
> > **Q5: "Why SDXL is much worse than SD in Table 1? DeepFloyd-IF 1024 images downsampled are strange. Is it possible to directly compare the 1024 resolution with the proposed method?"**
>
> **A5: 1)** As reported in Appendix Section F of the SDXL paper [b], its FID score is indeed worse than SDv1.5 and SDv2.1 on zero-shot MS-COCO dataset. Besides, we surpass all the baseline models including SDXL and SD series by a clear margin, in terms of both image quality metrics and the subjective user study. This can validate the superiority of our proposed method.
>
> **2)** Since the Ground Truth images of MS-COCO dataset are in resolution of 512x512, we have to downsample all the generated images to 512x512 for image-quality metrics evaluation. Note that the $\text{FID}$, $\text{KID}$, and $\text{FID}_\text{CLIP}$ are incomparable for images of different resolution, because they have to encode the same-resolution image features for distribution distance calculation.
>
> **3)** In terms of the comparisons in 1024x1024 resolution, we have already shown in the main paper with two aspects: **a)** All the presented visual comparisons against baselines are shown in 1024x1024 resolution. **b)** The user study is also conducted on generated images of 1024x1024 resolution. It can be clearly seen that our results are better than Deep-Floyd IF and multiple state-of-the-art baselines, generating more realistic images under high resolution.
>
> > **Q6: "About visual results from the first stage."**
>
> **A6:** Thank you so much for the precious advice! We have included more first-stage visual results in the Appendix Section A.10 "More First-Stage Generation Results". It can be seen that although the generation results are not as high-quality and realistic as those from the final two-stage pipeline, it still generates plausible humans with coherent structures.
>
> > **Q7: "Clarification of the intuition ‘Such monotonous images … low-frequency information.’."**
>
> **A7:** Thank you for pointing out this problem! We have polished to include the below clarifications in the Appendix Section A.12 "Detailed Intuition of Updated Noise Schedule":
>
> **1)** It is hard to finetune the Stable Diffusion to generate pure-color images. As shown in Figure 3(h) of the paper [c], we can not even overfit to a single solid-black image with the text prompt of "Solid black background". The main reason is that common diffusion noise schedules are flawed, which corrupts image incompletely when sampling $t=T$ at the training phase: $\mathbf{x}_T = \alpha_T \cdot \mathbf{x}_0 +  \sigma_T \cdot \mathbf{\epsilon}$, but $\alpha_T \neq 0, \sigma_T \neq 1$. Due to this reason, a small amount of signal is still included, which leaks the lowest frequency information such as the overall mean of each channel. In contrast, at the inference stage, the sampling starts from a pure Gaussian noise, which has a zero mean. Such train-test gap hinders SD from generating pure-color images.
>
> **2)** Similar to pure color images, the depth and surface-normal maps are visualized based on certain scheme, where its color and patterns are highly constrained. For example, the depth map is grey-scale image without colorful textures, and current estimators tend to infer similar depth values for each local patch. Therefore, the low frequency information of per-channel mean and standard deviation could be misused by network as shortcut for denoising, which harms the joint learning of multiple modalities (RGB, depth, and surface-normal). Motivated by this, we propose to enforce the zero-terminal SNR ($\mathbf{x}_T = 0.0 \cdot \mathbf{x}_0 + 1.0 \cdot \mathbf{\epsilon}$, that is, $\alpha_T = 0, \sigma_T = 1$) to fully eliminate low-frequency information at the training stage, so that we manage generate both RGB images and structural maps of high quality at the inference stage.
>
> > **Q8: "About the datasets."**
>
> **A8:** Thanks again for acknowledging that our dataset could benefit future research! We definitely would love to release the dataset, but are not able to commit to exact timelines at this point.

---

> ### Author Response · Authors · 2023-11-20
> **Response to Reviewer HmdV [Part 4/4]**
>
> > **Q9: "Does the input pose also contain the face landmarks or hand poses? Or just the body skeleton? If it's just skeleton, would adding hand poses helps with the generated details?"**
>
> **A9: 1)** No, the current input pose is just the body skeleton without face or hand keypoints. We choose such skeleton mainly for the fair comparison consideration: three controllable T2I baselines all support the body skeleton as condition, while some of them can not be directly adapted to more fine-grained keypoints like face and hands. Besides, body skeleton is comparatively easier to obtain than face and hand keypoints, which allows more flexible user control.
>
> **2)** You are totally correct that adding more detailed poses helps with the generation! At the early stage of experiment, we verified the effectiveness of giving additional face and hand keypoints as conditions, which could indeed facilitate better image quality with more fine-grained face and hand generation. But in the later experiments, since we have to align our setting with baselines, we do not include them as conditions. We sincerely hope that you could kindly understand this.
>
> > **Q10: "Ethical issues of Responsible research practice (e.g., human subjects, data release)."**
>
> **A10:** Thank you for pointing out the potential ethical issues!
>
> **1)** We are also aware of the potential negative impacts under the misuse of this technique. **The intention of this work is to benefit the research community and applications**. In view of this, we have provided our ethical considerations in Appendix Section A.6.
>
> **2)** We believe that the proper use of this technique will enhance the machine learning research and digital entertainment. We provide some additional measures to avoid negative social impacts:
>
> * We can add a visible or invisible watermark to the generated images to make it clear that they are not real photographs of real people. This can help prevent malicious users from using the images to deceive or manipulate others;
>
> * We can consider limiting access to the generated images to only authorized users, such as researchers or professionals in the field. This can help prevent the general public from misusing the images;
>
> * We can clearly state the terms of use for the generated images, including restrictions on how they can be used and distributed. This can help prevent malicious users from using the images for harmful purposes;
>
> * We can develop ethical guidelines for the use of the generated images, and require all users to adhere to these guidelines. This can help ensure that the images are used in a responsible and ethical manner.
>
> ****
>
> **References**
>
> [a] - Heusel et al. "GANs Trained by a Two Time-Scale Update Rule Converge to a Local Nash Equilibrium." NIPS 2017.
>
> [b] - Podell et al. "SDXL: Improving Latent Diffusion Models for High-Resolution Image Synthesis." arXiv:2307.01952, 2023.
>
> [c] - Lin et al. "Common Diffusion Noise Schedules and Sample Steps are Flawed." arXiv:2305.08891, 2023.
>
>
> ****
>
> Please don’t hesitate to let us know if there are any additional clarifications or experiments that we can offer!

---

> > ### Comment · Reviewer_HmdV · 2023-11-22
> >
> > Thanks for the great efforts and detailed response. It clarifies most of my concerns. As also pointed out by other reviewers, I agree that releasing the dataset would add significant value to this work.

---

> > > ### Author Response · Authors · 2023-11-23
> > > **Thanks for your comments!**
> > >
> > > Dear Reviewer HmdV:
> > >
> > > We are delighted to hear that your concerns are addressed! Many thanks again for your very constructive comments, which have helped us improve the quality of the paper significantly.
> > >
> > > We will try our best to make this work better, and make the contribution of this work to the community more significant!
> > >
> > > Best,
> > >
> > > Paper 350 Authors.

---

### Official Review · Reviewer_zFqJ · 2023-11-02

**Soundness:** 4 excellent
**Presentation:** 4 excellent
**Contribution:** 4 excellent
**Rating:** 10
**Confidence:** 3

**Summary:**

The paper introduces HyperHuman, a novel framework designed to address the challenge of generating hyper-realistic human images from text and pose inputs. It builds upon the foundations of diffusion models, improving upon the limitations of existing models like Stable Diffusion and DALL·E 2 that often fail to generate human images with coherent parts and natural poses. The key innovation is the integration of structural information across different granularities within a unified model to generate realistic images. The authors also curated a new dataset, HumanVerse, featuring a vast number of images with comprehensive annotations.

**Strengths:**

1. The Structure-Guided Refiner and improved noise schedule for eliminating low-frequency information leakage showcase thoughtful technical innovation.

2. The Latent Structural Diffusion Model's ability to simultaneously process and align RGB images, depth, and surface normals could result in more accurate and detailed images.

3. The paper achieves state-of-the-art performance in generating diverse and high-quality human images, which, if validated, marks a significant advancement.

4. The HumanVerse dataset's extensive size and detailed annotations can greatly benefit the generative model by providing a diverse range of training examples. Is valuable for the community.

**Weaknesses:**

I don't have major concerns, but it would be beneficial to discuss the following questions:

1. Given the model's architecture involves shared modules for different data modalities, could you elaborate on how the framework ensures the distinctiveness of modality-specific features? In particular, is there any mechanism within the model to prevent potential feature homogenization and maintain the integrity of the unique distributions associated with RGB, depth, and normal maps?

2. It would be insightful to understand how the model generalizes to unseen poses and whether there are specific pose complexities that present challenges.

3. The computational requirements for annotation and training are significant, given the use of multiple GPUs. Are there potential optimizations or simplifications that could maintain performance while reducing resource demands?

4. The model can generate photo-realistic and stylistic images. How does the model balance realism with artistic style variations, and could there be a more detailed explanation of how this balance is achieved?

5. Considering the rapid pace of advancement in this field, what are the authors' plans for updating or maintaining the model to keep up with emerging techniques and standards?

**Questions:**

Please see the Weaknesses.

---

> ### Author Response · Authors · 2023-11-20
> **Response to Reviewer zFqJ [Part 1/3]**
>
> We sincerely thank the reviewer for your insightful comments and recognitions to this work, especially for acknowledging that:
> 1) Our proposed framework is novel to address the challenge of realistic human generation;
> 2) We improve upon the limitations of existing models;
> 3) The Structure-Guided Refiner and improved noise schedule show thoughtful technical innovation;
> 4) The Latent Structural Diffusion Model can process and align multiple modalities for more accurate and detailed results;
> 5) We achieve state-of-the-art performance with significant advancement in diverse and high-quality human images;
> 6) The extensive HumanVerse dataset is diverse with detailed annotations to benefit the generation community.
>
> We have polished the paper, added the experiment results and made the clarifications in the revised version. Note that the following polishments have been made according to your advice:
>
> * The generation results on unseen and challenging pose are included in Appendix Section A.7 "Model Robustness on Unseen and Challenging Pose".
>
> * The discussions on potential computation optimizations for annotation and training; and experiment results on a much smaller-scale dataset are added to Appendix Section A.8 "Potential Optimization to Reduce Computation Cost".
>
> Thanks again for your very constructive comments, which have helped us improve the paper quality significantly! Below we would like to provide point-to-point responses to all the raised questions:
>
> > **Q1: "About the reasons that the proposed framework can 1) ensure the distinctiveness of modality-specific features, and 2) prevent potential feature homogenization and maintain the integrity of the unique distributions"**
>
> **A1:** Such modality-specific distinctiveness and unique distribution learning mainly derive from two aspects: the structural expert branch design, and the training target design.
>
> **1)** From the structural expert branch design perspective:
>
> * Each modality branch has its **own set of parameters**, containing conv_in+first DownBlock and last UpBlock+conv_out, which are the *closest* layers to the model input and output. In this way, the input from each modality can be encoded by modality-specific blocks. Then, after a series of shared modules, information is sufficiently interchanged to complement each other. Finally, the modality-specific decoders output noises of different modalities.
>
> * The skip-connections from the first DownBlock to the last UpBlock are separately connected for each modality, *i.e.*, the output of RGB branch’s DownBlock is skip-connected to RGB branch’s UpBlock, and similar for the depth and surface-normal branches. Therefore, the skip-connected features do not have the homogenization problem, and can be used to preserve modality-specific distinctiveness.
>
> **2)** From the training target design perspective:
>
> * As shown in Eq.4 of the main paper, we sample *different noises for each modality*, enforcing each branch to learn the modality-specific features to denoise the corresponding modality. Otherwise, if the model can not distinguish different modalities, it fails to converge to predict each modality’s noise correctly.
>
> > **Q2: "How the model generalizes to unseen poses? Whether there are specific pose complexities that present challenges?"**
>
> **A2: 1)** Thank you so much for the insightful suggestions! We further verify the model’s robustness and generalization ability on unseen and challenging pose of acrobatic pose. Specifically, we choose an acrobatic-related image from the Human-Art dataset [f], which is a highly challenging and rare pose unseen from the common human-centric images. The generation results and discussions are included in Appendix Section A.7 "Model Robustness on Unseen and Challenging Pose". It can be seen from Figure 6 that our model generalizes well to unseen poses. We also agree that generalization to highly rare/challenging poses is an open problem. We will explore this in future work.
>
> **2)** The MS-COCO 2014 validation dataset itself is zero-shot, which means all the images and human poses are unseen at the training phase. Besides, all the quantitative evaluations, qualitative results and visual comparisons are generated from the zero-shot MS-COCO, which shows that we can generalize well to unseen poses and prompts.
>
> **3)** Our dataset establishment process naturally guarantees the pose diversity of HumanVerse. Specifically, various human samples are curated, including different age groups, appearance, contexts (background scenes), poses, wearings, etc. As elaborated in the "Dataset Preprocessing" paragraph of main paper Section 4, even the partial human images such as clothing and limbs are preserved. This enables us to train an in-the-wild human generation model that is robust to unseen poses.

---

> ### Author Response · Authors · 2023-11-20
> **Response to Reviewer zFqJ [Part 2/3]**
>
> > **Q3: "The potential computation optimizations for annotation and training."**
>
> **A3:** Thank you for pointing out this interesting problem, which we believe can help our method adapt to different computational settings! We provide potential optimizations and simplifications from the training and annotation perspectives. All the discussions and experiment results are added to Appendix Section A.8 "Potential Optimization to Reduce Computation Cost".
>
> **1)** From the perspective of optimizing training:
>
> * We can change our models into a smaller diffusion backbone to save the training and memory cost, *e.g.*, Small SD (https://huggingface.co/segmind/small-sd) and Tiny SD (https://huggingface.co/segmind/tiny-sd) [a, b], which achieve on-par performance with Stable Diffusion, but lighter and faster in training and inference.
>
> * We can leverage some efficient parameter fine-tuning techniques like LoRA [c] and Adapter [d] to finetune the shared backbone with fewer parameters.
>
> * We can adopt some common engineering tricks to reduce memory consumption, *e.g.*, gradient checkpointing, gradient accumulation with smaller batch size, deepspeed model parallelism, lower floating point precision like fp16, efficient_xformers, etc.
>
> **2)** From the perspective of optimizing annotation:
>
> * Our efficient architecture design (only adding lightweight branches) can actually produce reasonable results with smaller dataset scale and fewer training iterations, capturing the joint distribution of RGB, depth, and surface-normal. Before the large-scale training, we first verify method effectiveness on a small-scale 1M subset, which is less than 3% of the HumanVerse full-set scale. In spite of this, we can still obtain good results with only 8 40GB A100 within one day, generating spatially aligned results for each modality. A generation sample is shown in Figure 7 of the Appendix Section A.8 "Potential Optimization to Reduce Computation Cost". Note that since this is an early-stage experiment, the pose conditioning and visualization are little bit different from the final version we have used. In spite of this, we manage to achieve simultaneous denoising of multiple modalities with a much smaller dataset scale.
>
> * The annotation overhead mostly comes from the diffusion-based image outpainting process (the "Outpaint for Accurate Annotations" paragraph of main paper Section 4), while the cost for depth and normal estimation is relatively low. Though facilitating more accurate pose annotations, it is not a mandatory step. Moreover, in the final evaluation process, we use the raw human pose without the help of outpainting, but can still achieve superior performance.
>
> > **Q4: "How does the model balance realism with artistic style variations, and could there be a more detailed explanation of how this balance is achieved?"**
>
> **A4: 1)** Currently, the balance between the photo-realism and artistic style variations comes from two aspects:
>
> * If we want to enforce the generation of either photorealistic or artistic styles, we can use text prompts as explicit guidance, *e.g.*, the prefix of "artistic renderings of …", "a DSLR photo of …", etc.
>
> * The finetune dataset domain influences the realistic-artistic balance a lot. For example, HumanSD [e] is trained on Human-Art dataset [f], which mostly consists of stylistic images. Therefore, they struggle to generate realistic images at the inference stage. In contrast, our dataset filtering process only requires the existence of partial human body, preserving both the photo-realistic and stylistic images for model training.
>
> **2)** This is a very insightful question ignored by most existing studies! We come up with a potential idea thanks to your inspiration:
>
> * We can label each image with a binary attribute of whether it is a photo-realistic one or a stylistic one. Such a condition can be taken as model input, similar to how we add the size conditioning in this work (the "Implementation Details" paragraph of main paper Section 5). In this way, we are free to adjust which style to generate at the inference stage. If such labeling is the extent of how photo-realistic or stylistic the image is, *i.e.*, a continuous value, we can even smoothly control the extent of photo-realism and artistic style variations. We will explore this interesting problem in future work.

---

> ### Author Response · Authors · 2023-11-20
> **Response to Reviewer zFqJ [Part 3/3]**
>
> > **Q5: "Plans for updating the model to keep up with emerging techniques and standards?"**
>
> **A5: 1)** We really appreciate your advice, and we are also envisioning a long-term progress in this important research problem! From our side, we are actively paying attention to the most up-to-date improvements in the general T2I domain to see if such insights can benefit our model, such as:
>
> * From the dataset perspective, Emu [g] finetunes on a carefully filtered small subset for quality tuning to improve visual results;
>
> * From the architecture perspective, Pixart-alpha [h] optimizes a pure transformer-based diffusion backbone for better efficiency;
>
> * From the caption perspective, recent papers like DALLE 3 [i] demonstrate that more detailed and accurate image captions are crucial.
>
> **2)** As an early attempt at in-the-wild human generation foundation model, we hope to
> pave the way for future research in this domain. We will continue to explore this problem, embrace new ideas, and try our best to make text-to-human, one of the most challenging sub-domains in T2I generation, a long-term development.
>
> ****
>
> **References**
>
> [a] - Bo-Kyeong et al. "On Architectural Compression of Text-to-Image Diffusion Models." arXiv:2305.15798, 2023.
>
> [b] - Bo-Kyeong et al. "BK-SDM: Architecturally Compressed Stable Diffusion for Efficient Text-to-Image Generation." ICML Workshop, 2023.
>
> [c] - Hu et al. "LoRA: Low-Rank Adaptation of Large Language Models." ICLR, 2022.
>
> [d] - Houlsby et al. "Parameter-Efficient Transfer Learning for NLP." ICML, 2019.
>
> [e] - Ju et al. "HumanSD: A Native Skeleton-Guided Diffusion Model for Human Image Generation." ICCV, 2023.
>
> [f] - Ju et al. "Human-Art: A Versatile Human-Centric Dataset Bridging Natural and Artificial Scenes." CVPR, 2023.
>
> [g] - Dai et al. "Emu: Enhancing Image Generation Models Using Photogenic Needles in a Haystack." arXiv:2309.15807, 2023.
>
> [h] - Chen et al. "PIXART-α: Fast Training of Diffusion Transformer for Photorealistic Text-to-Image Synthesis." arXiv:2310.00426, 2023.
>
> [i] - Betker et al. "Improving Image Generation with Better Captions." https://cdn.openai.com/papers/dall-e-3.pdf, 2023.
>
> ****
>
> Please don’t hesitate to let us know if there are any additional clarifications or experiments that we can offer!

---

### Author Response · Authors · 2023-11-20
**General Response**

We sincerely thank all the reviewers for your constructive feedbacks and recognitions to this work, especially for acknowledging the strengths of:
1) **Superior results with state-of-the-art performance** (Reviewer zFqJ, HmdV, 1Ma5, jWKR),
2) **Address an important problem to improve upon the limitations of existing models** (Reviewer zFqJ, HmdV, jWKR),
3) **Extensive dataset with detailed annotations notably benefits research community** (Reviewer zFqJ, HmdV, 1Ma5, jWKR),
4) **Generate high-quality human images in diverse scenarios** (Reviewer zFqJ, 1Ma5, jWKR),
5) **Novel framework with clear motivations and innovations** (Reviewer zFqJ, 1Ma5, jWKR),
6) **Simple yet effective with thoughtful technical innovation** (Reviewer zFqJ, HmdV, 1Ma5),
7) **Expert branches have additional ability to also denoise depth and surface-normal, interesting and well-study** (Reviewer zFqJ, HmdV, jWKR),
8) **Structural-Guided Refiner can better compose the predicted conditions and improve image quality** (Reviewer zFqJ, 1Ma5, jWKR),
9) **Extensive experiments and ablation studies to verify the effectiveness of each design** (Reviewer 1Ma5),
10) **Thorough evaluations and comparisons with multiple SOTA baselines** (Reviewer HmdV, 1Ma5),
11) **Well-written and easy to read** (Reviewer HmdV, jWKR).

****

We have polished the paper, added the experiment results, and made the clarifications in the revised version. Our manuscript is revised to include the following changes according to all the reviewers’ insightful comments, which have helped us improve the paper quality significantly! Note that all the polishments on the main paper and appendix are highlighted with **blue** text color for better visualization.

* We have included the generation results on unseen and challenging pose in Section A.7 "Model Robustness on Unseen and Challenging Pose" of the Appendix.

* We have added the discussions on potential computation optimizations for annotation and training; and experiment results on a much smaller-scale dataset in Section A.8 "Potential Optimization to Reduce Computation Cost" of the Appendix.

* We have included the unconditional generation results and analysis in Section A.9 "Model Performance without Input Pose" of the Appendix.

* We have added the jittered pose and animation experiment results and analysis in Section A.10 "Model Performance on Jittered Pose and Image Animation" of the Appendix.

* We have added the evaluation results for "Denoise RGB + Normal" ablation setting to Table 2 of the main paper.

* We have included more first-stage visual results in Section A.10 "More First-Stage Generation Results" of the Appendix.

* We have added the Clarifications of updated noise schedule in Section A.12 "Detailed Intuition of Updated Noise Schedule" of the Appendix.

* We have made the clarifications on pose processing and encoding details in Section A.13 "More Details on Pose Processing and Encoding" of the Appendix.

* We have added the inference requirements in Section 5 "Implementation Details" paragraph of the main paper.

* We have added the pose skeleton figures for all the qualitative comparisons in the Figure 13, 14, 15, 16 of Section A.13 "More Details on Pose Processing and Encoding" of the Appendix.

* We have polished the pipeline illustration in Figure 2 of the main paper to make it easier recognizable.

* We have included the modality-specific reconstruction results and analysis in Section A.14 "VAE Reconstruction Performance on Modality-Specific Input" of the Appendix.

Please don't hesitate to let us know of any additional comments on the manuscript or the changes.

---

### Author Response · Authors · 2023-11-21
**Welcome Further Response and Discussion**

Dear Reviewers:

We sincerely thank you again for your great efforts in reviewing this paper, especially for the recognitions to our work and the precious advice that has helped us improve the quality of this paper significantly!

We have polished the paper, added the experiment results, and made the clarifications in the revised version. **As the deadline for Author-Reviewer discussion is approaching**, we would like to use this opportunity to see if our responses are sufficient and if any concern remains. Please don't hesitate to let us know if there are further clarifications or experiments that we could offer. Thanks again for your time!

Best,

Paper 350 Authors.

---

### Meta-Review · Area_Chair_MheR · 2023-12-07

**Metareview:**

This work proposes to adapt stable diffusion for better human generation with new dataset finetuning as well as several techniques such as predicting depth and normals. All the reviewers lean towards accepting the work. Reviewers appreciated the well-written paper and very good results. Reviewers raised several clarification questions which are sufficiently addressed in the author responses. The reviewers did raise some valuable concerns that should be addressed in the final camera-ready version of the paper, which include adding the relevant rebuttal discussions and revisions in the main paper. The authors are encouraged to make the necessary changes to the best of their ability. As several reviewers suggested, it would be good if authors can release their dataset to propel further research in this space.

**Justification For Why Not Higher Score:**

Much of the contributions in the paper come with training on a new human dataset. Multiple reviewers asked for the dataset release, but authors did not confirm the data or model release suggesting that they are not willing to release the data, despite being one of the main contributions of the paper.

**Justification For Why Not Lower Score:**

All the reviewers are happy with the dataset and technical contributions in this paper.

---

### Decision · Program_Chairs · 2024-01-16

Accept (poster)